# Human DNA polymerase delta requires an iron–sulfur cluster for high-fidelity DNA synthesis

Stanislaw K. Jozwiakowski, Sandra Kummer, Kerstin Gari ⓘ

**Replication of eukaryotic genomes relies on the family B DNA polymerases Pol α, Pol δ, and Pol ε. All of these enzymes co-ordinate an iron–sulfur (FeS) cluster, but the function of this cofactor has remained largely unclear. Here, we show that the FeS cluster in the catalytic subunit of human Pol δ is coordinated by four invariant cysteines of the C-terminal CysB motif. FeS cluster loss causes a partial destabilisation of the four-subunit enzyme, a defect in double-stranded DNA binding, and compromised polymerase and exonuclease activities. Importantly, complex stability, DNA binding, and enzymatic activities are restored in the presence of proliferating cell nuclear antigen. We further show that also more subtle changes to the FeS cluster-binding pocket that do not abolish FeS cluster binding can have repercussions on the distant exonuclease domain and render the enzyme error-prone. Our data hence suggest that the FeS cluster in human Pol δ is an important co-factor that despite its C-terminal location has an impact on both DNA polymerase and exonuclease activities, and can influence the fidelity of DNA synthesis.**

## Introduction

Efficiency and fidelity of DNA replication determine genome stability and prevent premature ageing and cancer (Zeman & Cimprich, 2014). According to recent studies, more than 60% of mutations in human cancers are caused by replication errors (Tomasetti et al, 2017). In eukaryotes, DNA replication is largely dependent on the family B DNA polymerases Pol α, Pol δ, and Pol ε (Lujan et al, 2016) with Pol α being able to generate hybrid RNA–DNA primers to initiate DNA replication (Pellegrini, 2012), whereas the two most accurate eukaryotic DNA polymerases, Pol δ and Pol ε, are responsible for the bulk of nuclear DNA synthesis (Kunkel & Burgers, 2014). The accuracy of these enzymes is primarily determined by the stringent nucleotide selectivity of their DNA polymerase domains (Swan et al, 2009; Hogg et al, 2014). In addition, both enzymes are equipped with a 3′–5′ DNA exonuclease domain that provides a proofreading function and allows for the immediate correction of

DNA synthesis errors (Morrison et al, 1991). The importance of their proofreading function has been highlighted by studies with exonuclease-deficient *Pold1* and *Pole* knock-in mice that display a strong mutator and tumour-prone phenotype (Goldsby et al, 2001; Albertson et al, 2009). More recently, it was also reported that mutations in the proofreading domains of human Pol δ and ε predispose to colorectal and endometrial cancer and are associated with hypermutated tumours (Palles et al, 2013; Rayner et al, 2016). To date, The Cancer Genome Atlas lists 164 and 363 cancer-associated variants for *POLD1* and *POLE*, respectively (https://cancergenome.nih.gov), most of which have not been functionally characterised. Identifying error-prone variants of POLD1 and POLE and understanding the mechanisms that underlie their fidelity defects is, hence, also important in the context of cancer therapy (Nebot-Bral et al, 2017).

Whereas Pol ε synthesises primarily the leading strand, Pol δ is mostly responsible for the elongation of the nascent lagging strand (Pursell et al, 2007; Nick McElhinny et al, 2008; Georgescu et al, 2014). Human Pol δ is a heterotetramer comprising a catalytically active subunit (POLD1/p125), and three accessory subunits (POLD2/p50, POLD3/p66, and POLD4/p12) (Fig 1A). Stimulation of Pol δ by the replication clamp proliferating cell nuclear antigen (PCNA) is essential for processive lagging strand DNA synthesis (Stodola & Burgers, 2016). POLD1, POLD3, and POLD4 all contain a motif termed PCNA-interacting protein box that confers binding to PCNA (Acharya et al, 2011; Zhang et al, 2013).

Human POLD1 has sequence homology with other family B DNA polymerases from eukarya, archaea, and viruses (Nicolas et al, 2016) and shares their structural architecture encompassing an amino terminal domain (NTD), a 3′–5′ DNA exonuclease, a DNA polymerase and a carboxyl terminal domain (CTD) (Fig 1B). The CTD of POLD1 contains two highly conserved cysteine-rich motifs, CysA and CysB (Fig 1B). Several years ago, a study on yeast Pol δ demonstrated that CysA binds a $Zn^{2+}$ ion and constitutes an additional interaction site with PCNA that is required for the assembly of a stable Pol δ–PCNA complex on DNA (Netz et al, 2011). In contrast, the four invariant cysteines within CysB were shown to coordinate a [4Fe-4S] cluster that is essential for Pol δ complex assembly (Netz et al, 2011). Interestingly, the long-studied yeast strain *pol3-13* contains a single point mutation that causes a cysteine-to-serine change within the CysB motif (C1074S)

---

Institute of Molecular Cancer Research, University of Zurich, Zurich, Switzerland

Correspondence: s.k.jozwiakowski@live.com; gari@imcr.uzh.ch

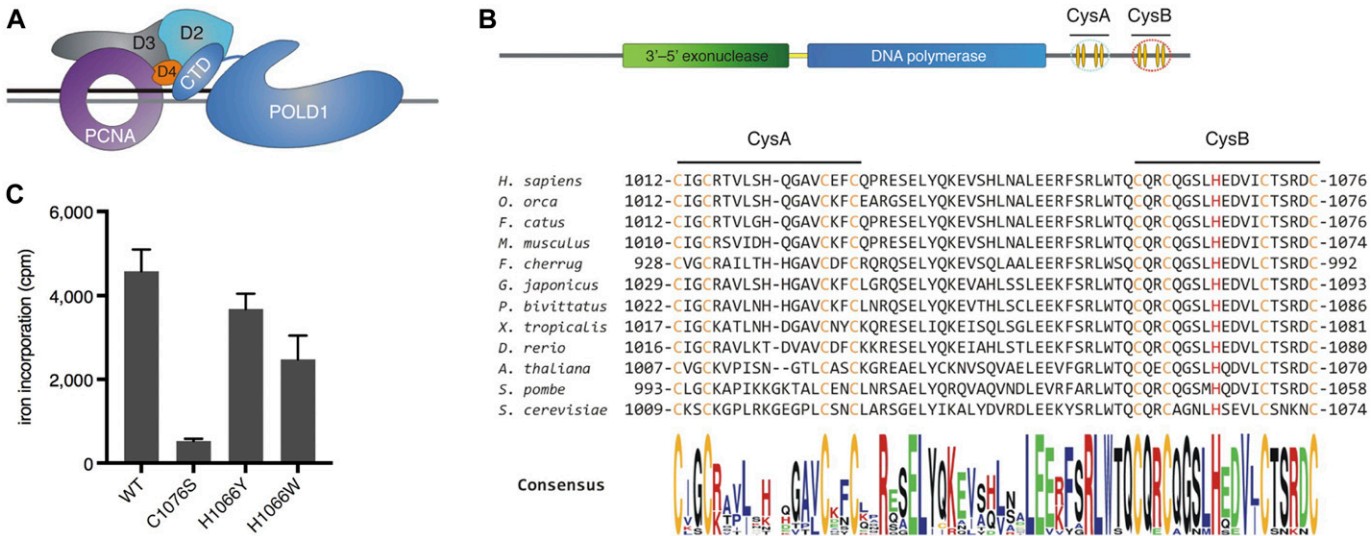

**Figure 1.  Human Pol δ coordinates an FeS cluster within CysB.**
**(A)** Schematic depicting Pol δ and PCNA on a DNA substrate. D2: POLD2, D3: POLD3, D4: POLD4, and CTD: C-terminal domain. **(B)** Schematic depicting linear arrangement of POLD1 domains (top). The sequence encompassing CysA and CysB from various species is aligned (bottom). Highlighted in orange are the invariant cysteines of CysA and CysB, in red the conserved histidine (H1066) residue within CysB. **(C)** Quantification of radioactive iron incorporation into wild-type and CysB-variant POLD1, as measured by liquid scintillation counting. Error bars depict standard deviations from three independent experiments. cpm, counts per minute.

(Giot et al, 1997), which is now known to compromise FeS cluster binding (Netz et al, 2011). *Pol3-13* is a temperature-sensitive strain and displays a strong DNA replication defect at restrictive temperatures, as well as irradiation sensitivity and defects in DNA repair even at permissive temperatures (Giot et al, 1997). Although it is defective in UV-induced mutagenesis, it displays an increased spontaneous mutation rate, which is largely dependent on the translesion DNA polymerase zeta (Pol ζ) (Stepchenkova et al, 2017).

The CTD of human POLD1 was more recently also shown to coordinate an FeS cluster (Baranovskiy et al, 2012); however, the role of this cofactor in human Pol δ was not further investigated. Here, we show that FeS cluster loss causes a partial destabilisation of the four-subunit enzyme, a defect in double-stranded DNA binding, and compromised polymerase and exonuclease activities. Importantly, all of these functions are restored in the presence of PCNA. We further show that also more subtle changes to the FeS cluster-binding pocket can affect the distant exonuclease domain and render the enzyme error-prone.

## Results

### POLD1 coordinates an FeS cluster via its CysB motif

Alignment of the POLD1 sequences of a variety of species reveals a high degree of conservation within the CTD (Fig 1B), with all eight invariant cysteines of the CysA and CysB motifs being conserved from human to yeast. Using a radioactive iron incorporation assay in *Sf9* insect cells (Fig S1A), we observed a strong reduction in iron incorporation when any of the four invariant cysteines of CysB were replaced with alanine within the CTD fragment (Fig S1B), suggesting that these residues are required for FeS cluster ligation. As in yeast (Netz et al, 2011), changing two cysteine residues at a time did not

further reduce iron incorporation (Fig S1B). Substitution of each of the four cysteine residues with glycine in full-length POLD1 led to a similar reduction in iron incorporation, although in this experimental setup, the third cysteine (C1071) appeared to contribute less to FeS cluster binding than the other ligating residues (Fig S1C). To reduce the impact of structural changes, we also generated a cysteine-to-serine variant (C1076S; CS) that displayed a similar reduction in iron incorporation as the corresponding cysteine-to-glycine or cysteine-to-alanine variants (Figs 1C and S1B and C).

In a number of FeS proteins, it has been shown that—apart from the cluster-ligating cysteines—other residues within the FeS cluster-binding pocket can potentially stabilise the cofactor, for example, protonable residues through hydrogen bonding (Bak & Elliott, 2013). In addition, the reactivity of cysteines can be modulated by charged amino acids in the vicinity (Britto et al, 2002). We were, therefore, interested in a highly conserved histidine located in the middle of CysB (Fig 1B). Predicting that this conserved residue could potentially influence FeS cluster binding, we also prepared a number of variants in which it was replaced with other amino acids. Of particular interest were the variants in which this residue was substituted with tyrosine (H1066Y; HY) and tryptophan (H1066W; HW) because they displayed FeS cluster binding that was reduced by about 20% in the case of the HY variant and by about 50% in the case of the HW variant (Fig 1C). This may suggest that replacing histidine 1,066 with tyrosine or tryptophan induces structural distortions in the CysB motif that lead to altered cysteine ligand geometry and reduced FeS cluster binding.

In conclusion, we show that human Pol δ coordinates an FeS cluster within its catalytic subunit. Replacing one of the four invariant cysteines of the CysB motif leads to a nearly complete loss of the FeS cluster. Our data further suggest that FeS cluster coordination is also affected by other residues within the FeS cluster-binding pocket.

## POLD1 assembles into a four-subunit structure in the absence of an FeS cluster

In yeast Pol δ, loss of the FeS cluster was shown to completely abolish the interaction of the catalytic subunit Pol3 with the two small subunits in vitro (Netz et al, 2011). To address whether the FeS cluster-binding pocket plays a role in the multi-subunit assembly of human Pol δ, we expressed N-terminally Flag-tagged *POLD1*, untagged *POLD2*, *POLD3*, and *POLD4* with or without *PCNA* in *Sf9* insect cells and performed a Flag-pull down (Fig 2A). In contrast to yeast Pol δ, multi-subunit assembly of human Pol δ seems to be largely independent of an FeS cluster because POLD1 was able to interact with all three subunits in the absence of an FeS cluster (Pol δ-CS) or upon alterations in the FeS cluster-binding pocket (Pol δ-HY/HW) (Fig 2A). Given that human Pol δ is a four-subunit polymerase (Liu et al, 2000), whereas yeast Pol δ is a three-subunit

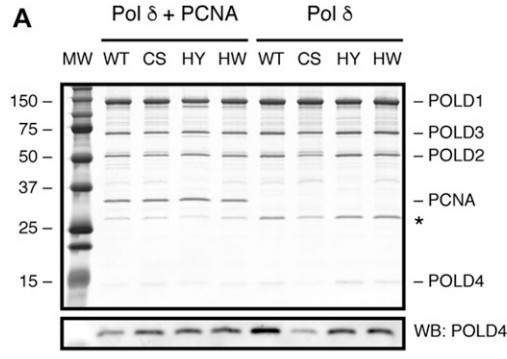

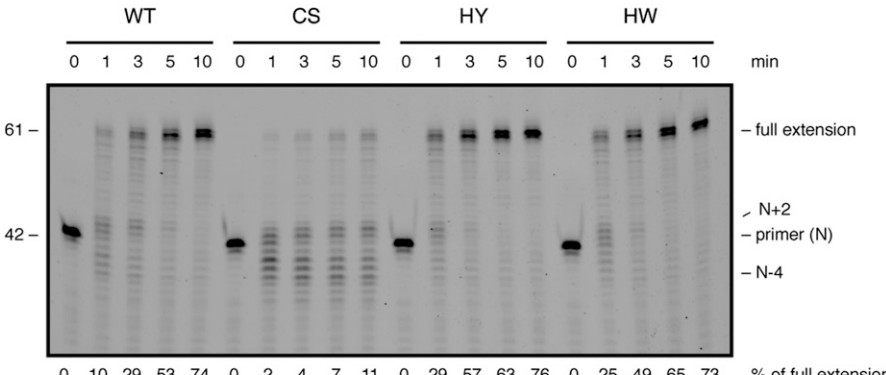

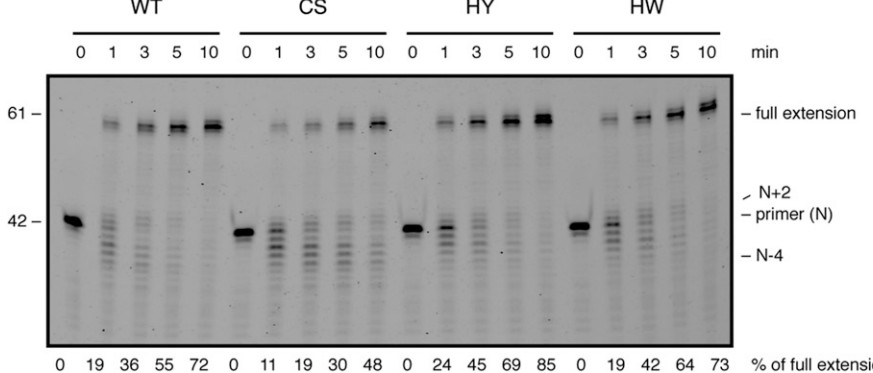

**Figure 2. FeS cluster loss affects DNA synthesis.**
**(A)** SDS–PAGE showing purified Pol δ in the presence (left) or absence (right) of PCNA. Asterisk denotes baculovirus PCNA that copurifies with Pol δ. **(B)** Scheme of primer extension assay. Grey circle indicates 5′-fluorescein amidite label. **(C, D)** Time-course analysis of primer extension with 2 nM of the indicated enzymes in the absence (C) or presence (D) of PCNA. Products were resolved on a denaturing polyacrylamide gel. MW, molecular weight.

polymerase, this discrepancy may reflect an intrinsic difference between the two species.

It should be noted, however, that the amounts of POLD2 and POLD4 subunits associated with Pol δ-CS were partially reduced when Pol δ was purified in the absence of PCNA (Fig 2A, right side of gel). Interestingly, when Pol δ was purified in the presence of PCNA, no difference between the variants could be observed (Fig 2A, left side of gel). Moreover, when Flag-purified samples were loaded on a Superdex S200 size exclusion column, Pol δ and Pol δ-CS—purified in the presence of PCNA—had very similar elution profiles (Fig S2A). These data suggest that loss of the FeS cluster causes a partial destabilisation of the four-subunit structure and that PCNA has a compensatory stabilising effect, presumably because of its ability to bind to three of the subunits.

### Pol δ requires an FeS cluster for efficient DNA synthesis

The above gel filtration analysis (Fig S2A) shows that our Flag-purified samples—apart from fully assembled Pol δ—also contain partially assembled subcomplexes and aggregated proteins (notably a substantial amount of PCNA is found in the void fractions). Nonetheless, to preclude a possible gradual oxidation of the FeS cluster during a prolonged purification procedure, we decided to limit the purification scheme to a one-step Flag-pull down. Considering that the ratio between fully assembled and partially assembled Pol δ is very similar in Pol δ and Pol δ-CS samples and that sub-assemblies of Pol δ—with the exception of a three-subunit assembly of POLD1, POLD2, and POLD3 that we investigate further in a later paragraph—have only residual enzymatic activities, such Flag-purified Pol δ variants can be compared among one another.

To investigate DNA synthesis by Pol δ upon alterations in the FeS cluster-binding pocket, we then used a time-resolved primer extension assay (Fig 2B). When purified without PCNA, Pol δ-CS displayed a substantial DNA polymerisation defect with only 11% of primers being fully extended (Fig 2C). Importantly, a significant portion of the primers was extended only up to two nucleotides (N + 2) or degraded up to four nucleotides (N − 4). In contrast, Pol δ-HY/HW displayed robust DNA synthesis that was comparable with wild-type Pol δ with up to 75% of primers being fully extended (Fig 2C). Surprisingly, Pol δ-CS—purified in the absence of PCNA—displayed full primer extension upon addition of increasing amounts of purified PCNA (Fig S2B and C). Likewise, when purified in the presence of PCNA, Pol δ-CS was able to fully extend a substantial amount of primers (Fig 2D), even though it remained the least efficient enzyme with only 50% of primers reaching full extension—in contrast to 75% by the wild-type enzyme and Pol δ-HY/HW.

These findings are consistent with the observed partial destabilisation of the Pol δ-CS four-subunit structure and a stabilising role of PCNA (Fig 2A). To further investigate this destabilisation, we challenged the different CysB variants of Pol δ—purified in the presence of PCNA—by heat treatment at 55°C (Fig S3A). Whereas the wild-type enzyme and Pol δ-HY/HW were gradually inactivated over 5 min of incubation at 55°C, Pol δ-CS lost its enzymatic activity after only 1 min of heat treatment (Fig S3B). We speculate that the lower thermal resistance observed for the CS variant is caused by a more rapid de-oligomerisation of the four-subunit structure. Alternatively, given that DNA synthesis by Pol δ-CS is highly dependent on

PCNA, it is also possible that Pol δ-CS is particularly affected by the dissociation of PCNA, whereas the wild-type enzyme and the histidine variants can still efficiently synthesise DNA in the absence of PCNA (Fig 2C).

Taken together, our data demonstrate that loss of the FeS cluster in Pol δ causes a partial destabilisation of the four-subunit enzyme and a pronounced DNA polymerisation defect, both of which can be alleviated by PCNA.

### The FeS cluster binding–deficient variant of human Pol δ has a dsDNA-binding defect

To perform efficient DNA synthesis, all replicative DNA polymerases must correctly recognise and stably bind to the primer–template substrate. This ability is not only essential for the correct positioning of the primer–template junction in the DNA polymerase domain but also determines the processivity of these enzymes (Rothwell & Waksman, 2005). To address whether the DNA polymerisation defect of Pol δ-CS stems from incorrect binding to its DNA substrate, electrophoretic mobility shift assays (EMSAs) were used.

We first tested a classical primer–template substrate and found that all CysB variants were able to bind to this substrate, both in the absence and presence of PCNA (Fig 3A and B). Interestingly, at higher protein concentrations, two distinct shifted bands were discernible that may represent the binding of one enzyme (lower band) and simultaneous binding of two enzymes (higher band) to the DNA substrate. Because Pol δ contacts the primer–template substrate both in the double-stranded (ds) region (with the thumb and C-terminal domains of POLD1) and the single-stranded (ss) region (with the N-terminal and exonuclease domains of POLD1) (Swan et al, 2009), we reasoned that one of the enzymes could be bound correctly at the ds/ssDNA transition site, whereas the other could be bound in the unoccupied dsDNA region of the substrate. In comparison with the wild-type enzyme and the HY/HW variants, Pol δ-CS produced such a super-shift to a lesser degree, which prompted us to test whether Pol δ-CS may be deficient in binding to dsDNA.

Indeed, Pol δ-CS, but not Pol δ-HY/HW, was greatly impaired in binding to a dsDNA probe when purified without PCNA (Fig 3C). Interestingly, as for DNA synthesis, copurification of Pol δ-CS with PCNA alleviated this DNA binding defect (Fig 3D). Although dsDNA itself is not a natural substrate for Pol δ, the contact of POLD1 with dsDNA via its thumb and C-terminal domains is required for the correct binding of a primer–template substrate. The fact that in our experimental conditions, Pol δ-CS did not have a discernible defect in primer–template binding could be explained by a compensatory effect of the N terminus of POLD1. In agreement with this notion, Pol δ-CS did not display any defects in binding to a single-stranded DNA substrate (Fig 3E and F).

In conclusion, our EMSAs demonstrate that loss of the FeS cluster causes a defect in dsDNA binding that can be alleviated by the presence of PCNA. Impaired binding to dsDNA is likely to interfere with correct positioning and translocation of Pol δ on its DNA substrate and could, hence, explain the DNA synthesis defect observed with Pol δ-CS (Fig 2C).

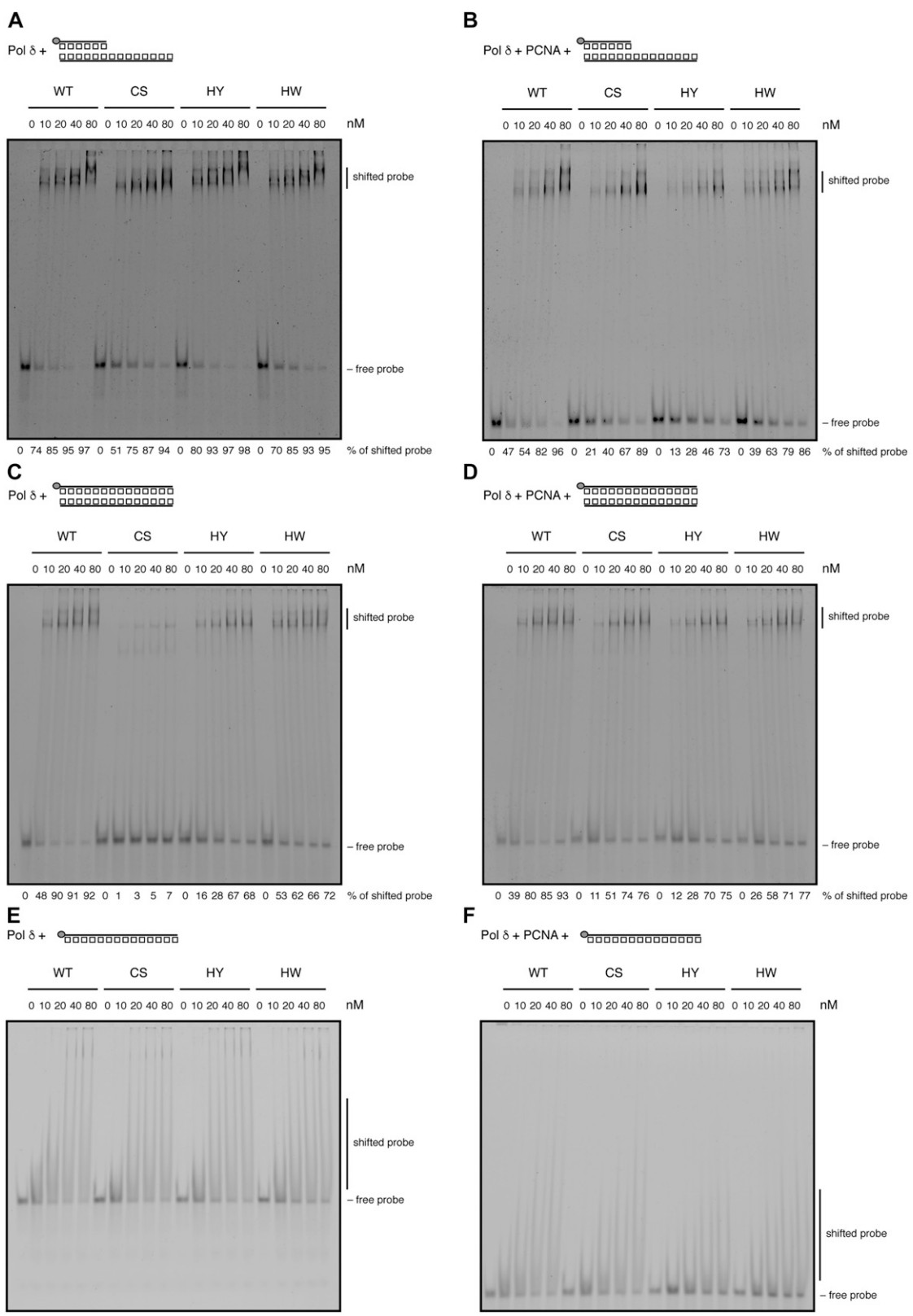

**Figure 3. FeS cluster loss affects binding to dsDNA.**
**(A, B)** DNA binding to primer–template junctions was analysed by EMSA with increasing amounts of Pol δ in the absence (A) or presence (B) of PCNA. **(C, D)** DNA binding to dsDNA was analysed with increasing amounts of Pol δ in the absence (C) or presence (D) of PCNA. **(E, F)** DNA binding to ssDNA was analysed with increasing amounts of Pol δ in the absence (E) or presence (F) of PCNA. Grey circle indicates 5′-fluorescein amidite label.

## Pol δ requires an FeS cluster for efficient DNA exonuclease activity

One of the key features of replicative DNA polymerases is their ability to correct their own errors by 3′–5′ exonuclease proofreading (Bebenek and Ziuzia-Graczyk, 2018). Work from Marietta Lee's laboratory suggests that in response to a variety of agents that cause DNA damage or DNA replication stress, the p12 subunit of human Pol δ gets rapidly degraded (Zhang et al, 2007). Moreover, the three-subunit complex that lacks p12 (Pol δ3) displays a higher exonuclease, but lower polymerase, activity in vitro than the four-subunit complex (Pol δ4) (Meng et al, 2009), which would be in line with a role for Pol δ3 under conditions of DNA replication stress when proofreading is likely being favoured over rapid DNA synthesis.

To investigate the influence of the FeS cluster on Pol δ's exonuclease activity, we purified all POLD1 variants in the context of the three- and four-subunit enzymes in the absence or presence of PCNA. When purified as part of the four-subunit enzyme—with or without PCNA—all three CysB variants displayed a reduced

exonuclease activity, as compared to the wild-type protein (Fig 4A–C). This partial defect in exonuclease activity was also observed when the CysB variants were purified as part of the three-subunit enzyme (Fig 4D–F). Strikingly, although Pol δ-HY/HW were fully proficient in DNA synthesis, they displayed defects in exonuclease activity, which were comparable with the ones observed with Pol δ-CS.

Taken together, these results suggest that alterations in the FeS cluster-binding motif of Pol δ—even if they do not, or only partially, affect FeS cluster binding—can cause defects in exonuclease activity.

## Pol δ requires an FeS cluster for high-fidelity DNA synthesis

Given the reduced exonuclease activity that we observed upon alterations in the FeS cluster-binding motif, we next used a plasmid-based *lacZα* forward mutation assay (Jozwiakowski & Connolly, 2009, Keith et al, 2013) to address whether the FeS cluster is required for DNA replication fidelity (Figs 5A and S4). Because Pol δ-CS was less

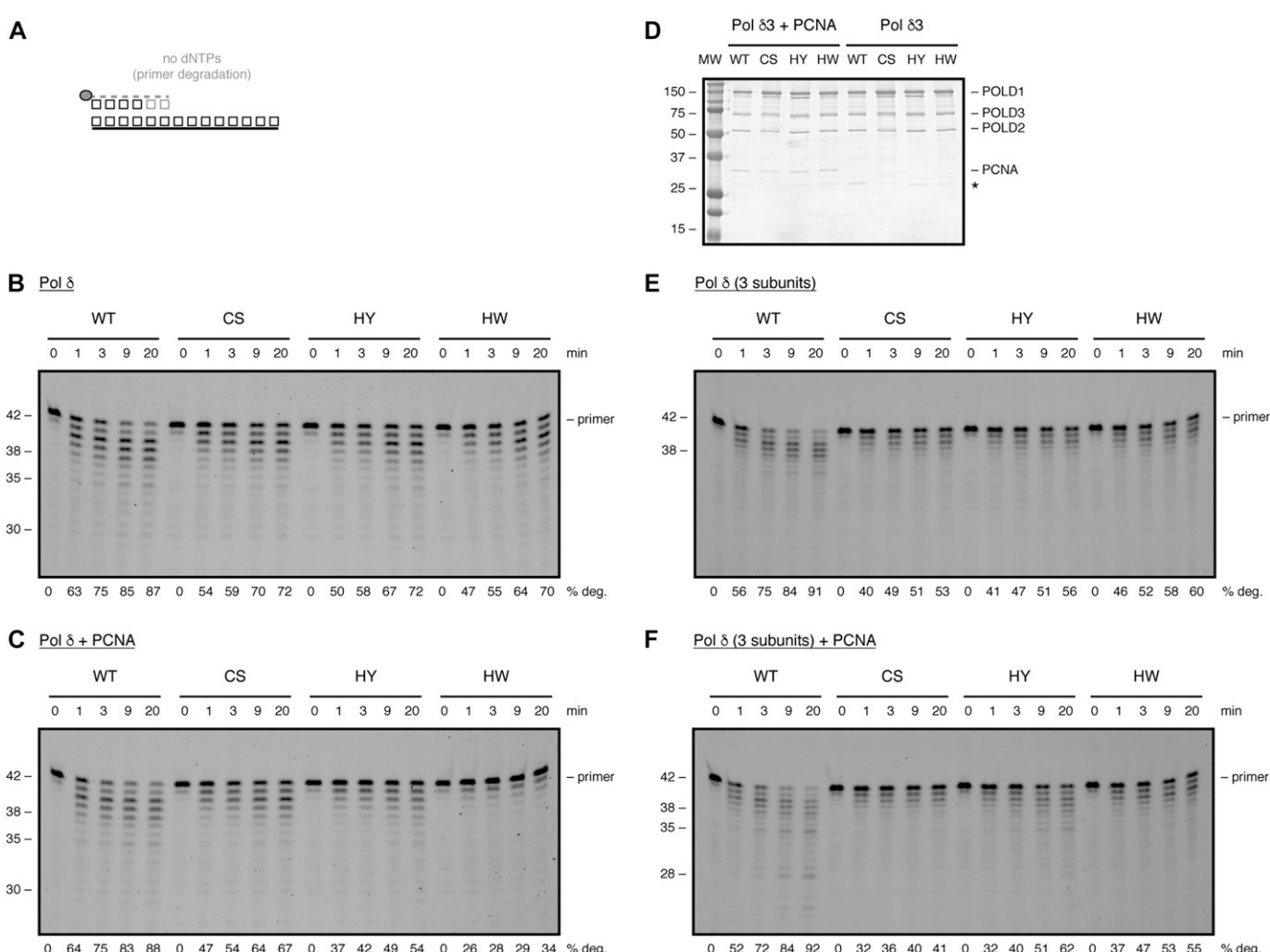

**Figure 4.   The FeS cluster has an impact on the exonuclease activity of Pol δ.**
**(A)** Scheme of exonuclease assay. Grey circle indicates 5′-fluorescein amidite label. **(B, C)** Time-course analysis of exonucleolytic degradation with 2 nM of the indicated Pol δ variants in the absence (B) or presence (C) of PCNA. **(D)** SDS–PAGE showing purified Pol δ3 in the presence (left) or absence (right) of PCNA. MW: molecular weight. **(E, F)** Time-course analysis of exonucleolytic degradation with 2 nM of the indicated Pol δ3 variants in the absence (E) or presence (F) of PCNA. Products were resolved on a denaturing polyacrylamide gel.

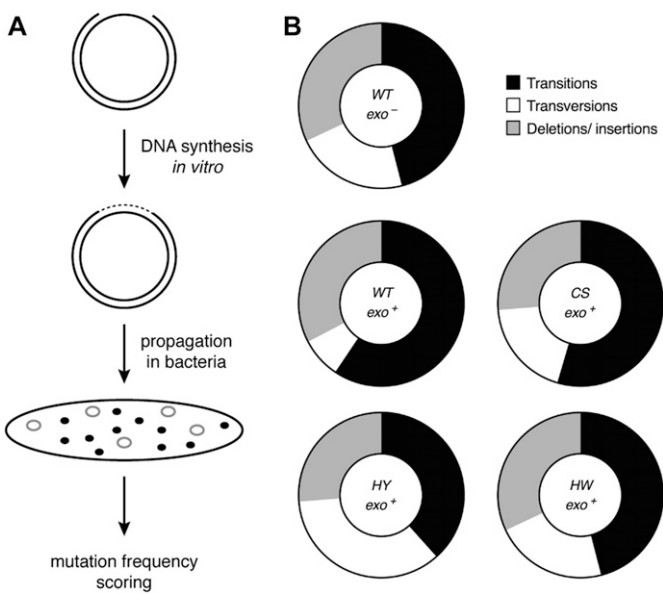

**Figure 5. The FeS cluster impacts on the fidelity of DNA replication.**
**(A)** Scheme of plasmid-based *LacZα* forward mutation assay. **(B)** Graphical depiction of percentage of transitions (black), transversions (white), and deletions/insertions (grey) caused by the indicated variants of Pol *δ*–PCNA complexes.

efficient in DNA synthesis even in the presence of PCNA (Fig 2D), we used a modified plasmid (pSJ4-*lacZα*) that contains only a 64-nucleotide-long gap to ensure completion of the gap-filling reaction. To assess whether the gapped pSJ4 plasmid can be used to study Pol *δ*, we started with fidelity measurements of the wild-type complex (WT exo⁺) and an exonuclease-deficient variant (WT exo⁻) in the absence of PCNA as a reference point (Table S1). Wild-type Pol *δ* inserted less than one error per 150,000 nucleotides polymerised, which is in agreement with previous data using human Pol *δ* purified from *E. coli* and a M13mp2 *lacZα* forward mutation assay (Schmitt et al, 2009). For the exonuclease-deficient variant of Pol *δ*, however, we observed only a 2.5-fold decrease in fidelity as compared with the proofreading-proficient variant of Pol *δ*, whereas Schmitt and colleagues had observed greater than 10-fold decrease in fidelity (Schmitt et al, 2009). Although this discrepancy can be explained by a number of factors,

including enzyme purification, the DNA substrate used, and the conditions of the gap-filling reaction, it suggests that our assay may be suboptimal for absolute fidelity measurements. However, because our aim was to compare wild-type Pol *δ* with CysB variants, rather than to measure absolute values, we deemed this assay nevertheless suitable for our purposes. To be able to compare the fidelity of all CysB variants, we performed the reactions in the presence of PCNA because Pol *δ*-CS was hardly able to synthesise DNA in the absence of PCNA (Fig 2C). As this setup required the presence of a clamp loader and an excess of ATP, we observed an overall decrease in fidelity (Table 1), as compared with the reactions without PCNA (Table S1), which is in agreement with a previous study, in which PCNA lowered the fidelity of calf thymus Pol *δ* (Mozzherin et al, 1996).

Remarkably, in our experimental setup, all CysB variants displayed a 2- to 2.5-fold reduced fidelity, as compared with the wild-type enzyme, which emulated the threefold reduction in fidelity observed with the exonuclease-deficient variant (Table 1). Moreover, whereas the percentage of deletions was comparable for all enzymes, the ratio of transversions over transitions increased sharply for the CysB variants, as compared with the wild-type complex (Figs 5B and S5 and Table S2). Interestingly, this mutation distribution is reminiscent of the mutation signature of the exonuclease-deficient variant (Figs 5B and S5 and Table S2). The observed tendency is also consistent with previous fidelity studies that demonstrate that transversions are more efficiently proofread by replicative DNA polymerases and that an increase in transversions is often correlated with deficient proofreading (Goodman et al, 1993; Kunkel & Bebenek, 2000).

Taken together, our data suggest that defects in exonuclease activity due to alterations in the FeS cluster-binding motif confer an error-prone phenotype and that the correct coordination of an FeS cluster is, hence, required for the ability of human Pol *δ* to perform high-fidelity DNA synthesis.

### The FeS cluster has an influence on the balance between DNA polymerisation and exonuclease activity

So far, no crystal structure is available for human Pol *δ* or its catalytic subunit POLD1. In an attempt to understand how alterations in the FeS cluster-binding pocket can have far-reaching impact on the catalytic

**Table 1. Error rates of exonuclease-deficient (WT exo⁻) or exonuclease-proficient (WT/CS/HY/HW exo⁺) Pol *δ* variants in the presence of PCNA in a pSJ4-*lacZα* forward mutation assay.**

| Pol *δ* + PCNA | Total number of colonies[a] | Number of white mutants | Corrected mutant frequency[b] | Error rate[c] |
|---|---|---|---|---|
| WT exo⁻ | 14,875 | 85 | $5.5 \times 10^{-3}$ | $1.2 \times 10^{-4}$ |
| WT exo⁺ | 20,245 | 40 | $1.8 \times 10^{-3}$ | $4.1 \times 10^{-5}$ |
| CS exo⁺ | 18,176 | 72 | $3.8 \times 10^{-3}$ | $8.6 \times 10^{-5}$ |
| HY exo⁺ | 20,153 | 68 | $3.2 \times 10^{-3}$ | $7.2 \times 10^{-5}$ |
| HW exo⁺ | 20,814 | 97 | $4.5 \times 10^{-3}$ | $1.0 \times 10^{-4}$ |

[a]The fidelity of each polymerase variant was determined in three separate experiments. The aggregated numbers are given.
[b]Mutant frequency equals: (number of white colonies/total number of colonies) – background mutant frequency. A background mutant frequency of $5.5 \times 10^{-6}$ was used for gapped pSJ4.
[c]Error rate is the number of mistakes made per base incorporated. The corrected mutant frequency was converted to error rate as previously described (Keith et al, 2013). An expression frequency (P) of 0.3 was used. Because of the limited amount of sequencing data, a set Ni/N value of 1 was used and the number of detectable sites (D) was the sum of the determined base substitutions plus insertions/deletions, that is, 145 in pSJ4.

activities of Pol δ, we generated a 3D model of POLD1 based on the available structures of the Pol3 and Pol1 subunits of yeast Pol δ (PDB: 3IAY) and Pol α (PDB: 5EXR), respectively (Swan et al, 2009; Baranovskiy et al, 2016). In our model (Fig 6A), the CTD forms a bundle of helices and is connected via a flexible linker to the thumb subdomain. The CTD is positioned above, and orientated in parallel to, the DNA axis. This positioning seems to be most logical, as it orients both the zinc-binding CysA motif and the PCNA-interacting protein box towards the plane of the PCNA ring. Moreover, in this arrangement, the CTD can serve as a scaffold to assemble the accessory subunits POLD2, POLD3, and POLD4 around dsDNA. In this orientation, the FeS cluster-binding CysB motif is located next to the flexible linker and points towards the palm subdomain. Interestingly, low-resolution structures of yeast Pol δ and Pol ε show an elongated shape, in which the globular catalytic domains are connected via a flexible linker to the C-terminal parts of the enzymes and their accessory subunits (Jain et al, 2009; Asturias et al, 2006), suggesting that the flexible linker could be required for the proper alignment of the enzymes on the DNA substrate. Based on the proximity of the FeS cluster to the flexible linker in our model, it seems conceivable that already small structural changes in the FeS cluster-binding pocket may be able to influence the conformational flexibility of the linker and—by doing so—affect the alignment of Pol δ on the DNA substrate and possibly the balance between DNA polymerase and exonuclease activities.

To test whether the equilibrium between the two catalytic activities of Pol δ is affected by alterations in the FeS cluster-binding motif, we carried out fixed-time primer extension assays in the presence of increasing amounts of dNTPs ranging from 0.01 to 100 μM (Figs 6B and S6). Wild-type Pol δ and Pol δ-HY/HW—purified in the presence of PCNA—displayed a comparable primer extension rate with quantifiable amounts of fully extended primers starting to accumulate at dNTP concentrations of 0.5–1 μM (Fig S6). Interestingly, however, the concentrations of dNTPs at which the enzymes started switching from exonuclease to DNA polymerisation mode were significantly divergent with the wild-type enzyme requiring 0.2 μM, whereas 0.02 μM were sufficient for Pol δ-HY/HW (Figs 6B and S6).

In striking contrast to the other complexes, Pol δ-CS needed at least 10 μM of dNTPs to generate quantifiable amounts of fully extended primers, and 5 μM of dNTPs to show more efficient DNA synthesis than degradation (Figs 6B and S6). Importantly, a substantial amount of primers (36–63%) was not processed at all and no progressive increase in the DNA polymerisation rate was observed across the range of dNTP concentrations.

Taken together, our data indicate that alterations in the FeS cluster-binding pocket of Pol δ that lead to a loss of its FeS cluster negatively affect both catalytic activities. Importantly, also more subtle distortions in the FeS cluster-binding pocket can have repercussions on the catalytic activities of Pol δ. Notably, replacement of the positively charged histidine 1,066 with a bulky aromatic residue seems to tip the balance between the two activities toward DNA polymerisation.

## Discussion

FeS clusters are ancient and evolutionary conserved cofactors with various functions in all kingdoms of life (Brzoska et al, 2006).

As redox-active entities, they can adopt redox potentials over a wide range (Meyer, 2008) and are best known for their function in the respiratory chain, where their redox activity is used for electron transport across the mitochondrial membrane (Brzoska et al, 2006). In recent years, a surprisingly large number of enzymes involved in DNA replication and repair have been found to be clients of the cytoplasmic FeS assembly (CIA) machinery (Gari et al, 2012; Stehling et al, 2012) and to coordinate an FeS cluster, including DNA primase and all members of the family B DNA polymerases in yeast (Klinge et al, 2007; Netz et al, 2011). Although a recent study has shown that yeast Pol δ is redox-active when bound to DNA (Bartels et al, 2017), it is so far largely unclear whether the redox activity of FeS clusters plays a role in the context of DNA replication.

Here, we provide evidence that the FeS cluster in human Pol δ has an important structural and functional role. In contrast to the situation in yeast (Netz et al, 2011), loss of the FeS cluster does not prevent the assembly of human Pol δ but causes a partial destabilisation of the multi-subunit structure that can be completely overcome by the presence of PCNA. Moreover, upon loss of the FeS cluster, we observe a strong DNA synthesis defect that correlates with an inability of the enzyme to bind to dsDNA, whereas binding to a primer–template substrate appears unaltered. Based on our structural model of POLD1, we speculate about a scenario (Fig 7) in which loss of the FeS cluster could lead to structural aberrations in the FeS cluster-binding pocket that cause the misalignment of the thumb subdomain and result in the inability of the complex to grip stably to, and to translocate along, dsDNA. This hypothetical scenario would be consistent with our observation that the presence of PCNA can alleviate both DNA synthesis and DNA-binding defects because PCNA is known to stabilise Pol δ on DNA and—by doing so—to stimulate processivity of DNA synthesis (Stodola & Burgers, 2016). In contrast to a loss of the FeS cluster, more subtle changes in the FeS cluster-binding pocket do not seem to impair DNA binding and DNA synthesis.

Interestingly, despite these differences in DNA binding, all three CysB variants investigated here display similar defects in exonucleolytic activity. There are, however, substantial differences between the three variants that become apparent in the dNTP concentration-dependent primer degradation-to-extension switch assay (Fig 6B). While loss of the FeS cluster—and as a consequence presumably a collapse of the binding pocket—reduces the ability of Pol δ-CS to switch to either DNA polymerase or exonuclease mode, the more subtle alterations in the FeS cluster-binding pocket of Pol δ-HY/HW seem to preferentially induce the DNA synthesis mode. The FeS cluster-binding pocket may, hence, be able to affect the conformational switch between the polymerase and exonuclease sites, a so far poorly understood process. Interestingly, the thumb subdomain was recently proposed to be a key regulator for enzyme translocation in a viral family B DNA polymerase (Ren, 2016), suggesting that it may be involved in the switching between DNA polymerase and exonuclease active sites. Because the FeS cluster-binding pocket in human Pol δ is located in the flexible linker region that connects the thumb subdomain with the C-terminal dsDNA-binding part of the enzyme, it would be in a prime position to affect domain switching. However, further structural and functional studies will be necessary to reveal the

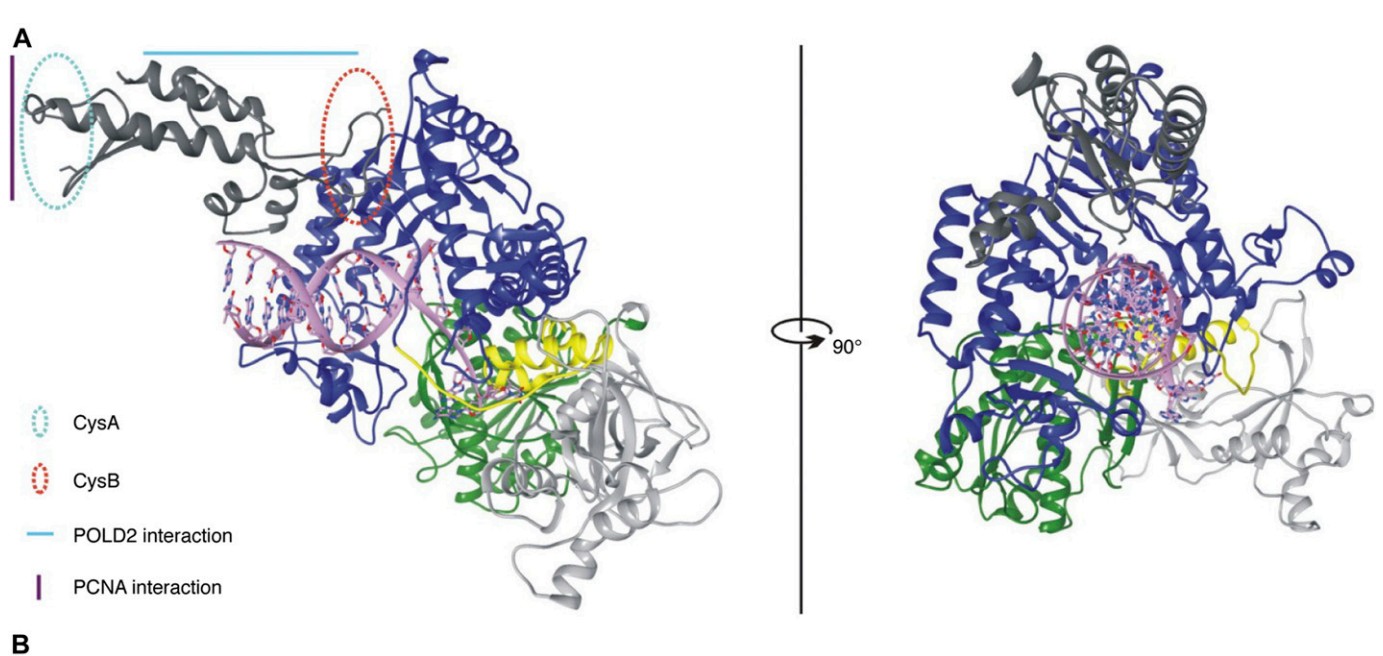

## B

Pol δ (WT) + PCNA

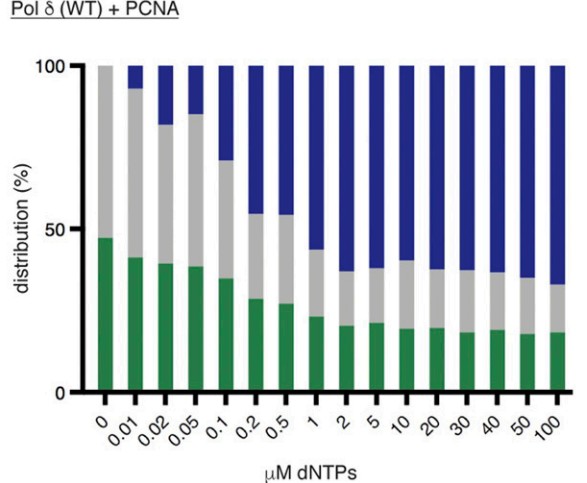

Pol δ (CS) + PCNA

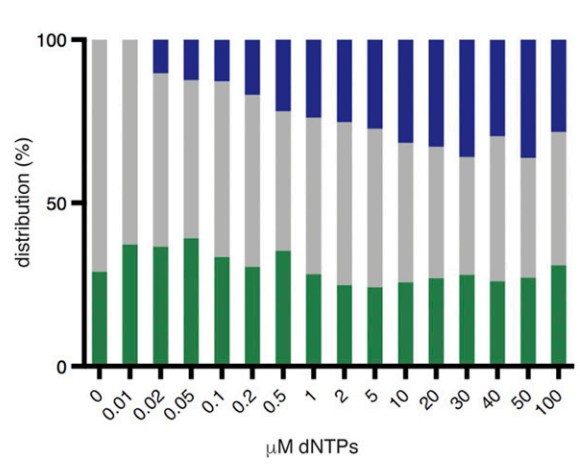

Pol δ (HY) + PCNA

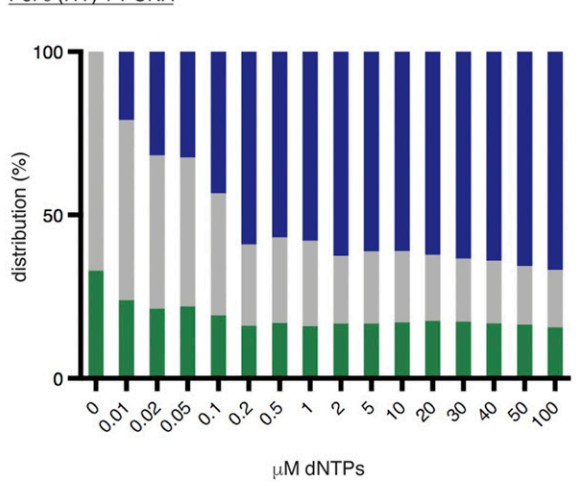

Pol δ (HW) + PCNA

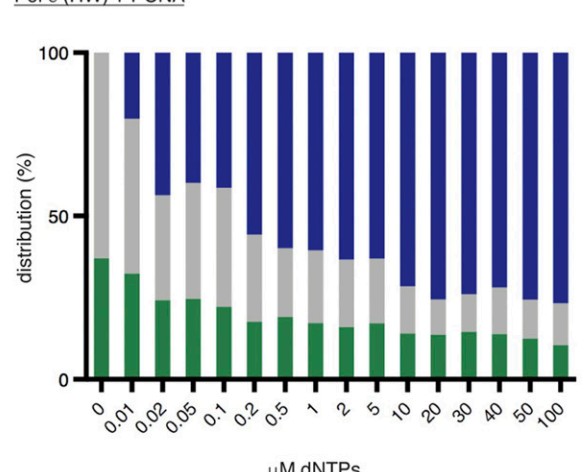

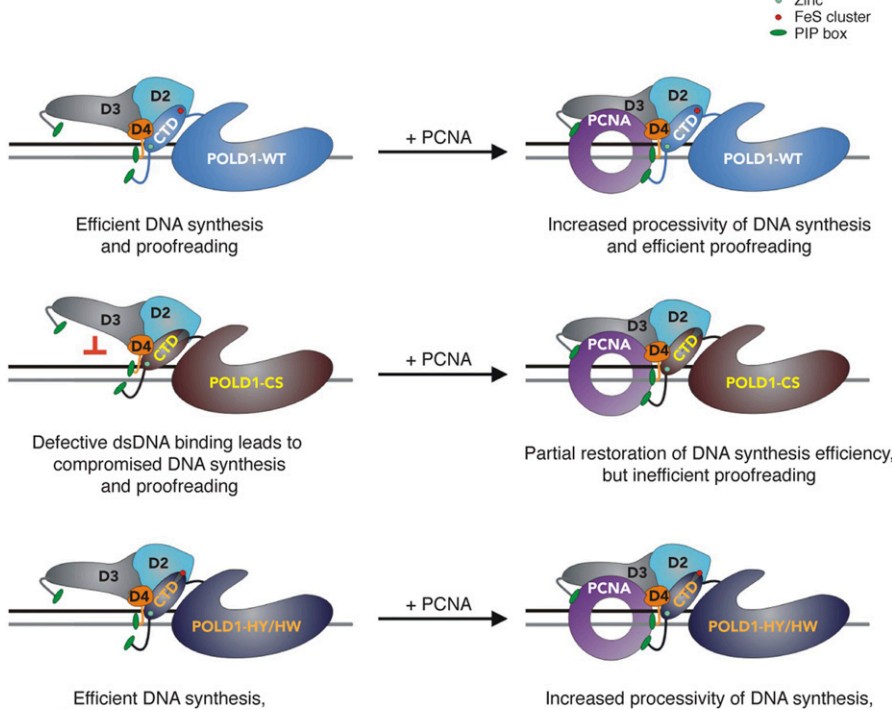

**Figure 7.  Hypothetical model.**
See the Discussion section for details.

Legend: Zinc · FeS cluster · PIP box

exact mechanism that controls the exonuclease-to-polymerase switch in human Pol δ.

In a highly speculative scenario, the FeS cluster could potentially influence the equilibrium between DNA synthesis and exonucleolytic degradation to enable high-fidelity DNA synthesis during unperturbed replication. During conditions of oxidative stress, however, the FeS cluster may be oxidised and as a consequence rapidly lost, which would slow down DNA synthesis and lower proofreading activity. Such a switch from a replicative to a repair synthesis mode could allow Pol δ to finish already initiated Okazaki fragments in a slow and potentially error-prone manner under DNA damage conditions. In line with this idea, there is a growing body of evidence that involves Pol δ in the tolerance of UV-induced DNA lesions (Hirota et al, 2015). It has remained unclear, however, how exonuclease-proficient Pol δ could bypass UV lesions efficiently, as its exonuclease activity acts as a strong kinetic barrier that needs to be attenuated for lesion bypass (Khare & Eckert, 2002).

Interestingly, mutations in *POLD1* are—apart from colorectal and endometrial cancer (Rayner et al, 2016)—also associated with a rare human disorder termed mandibular hypoplasia, deafness, progeroid features, and lipodystrophy (MDPL) syndrome (Reinier et al, 2015). In most cases, MDPL is linked to mutations that inactivate DNA polymerase function and modulate the exonuclease activity of human Pol δ (Weedon et al, 2013), but two very recent studies describe patients with point mutations in the CysB motif (Ajluni et al, 2017; Elouej et al, 2017). Although the resulting pathogenic POLD1 variants (E1067K and I1070A) need to be investigated, our data allow us to predict that they are likely to have compromised DNA polymerase and/or exonuclease activities and confer an error-prone phenotype because of structural alterations in the FeS cluster-binding pocket.

# Materials and Methods

### Cloning and baculovirus generation

Codon-optimised *POLD1*, *POLD2*, *POLD3*, and *POLD4* sequences were purchased for expression in *Sf9* insect cells (Gen9). *POLD1* was cloned into the GATEWAY entry vector pDONR221 (Invitrogen) and used as a template for the generation of the C-terminal domain fragment (CTD, 900–1,107 aa). *POLD2*, *POLD3*, and *POLD4* cDNAs were ordered such that their cDNAs were separated by a sequence coding for the self-cleaving T2A peptide (*POLD2-T2A-POLD3-T2A-POLD4*). This cassette was also cloned into pDONR221 and served as a template for the generation of *POLD2-T2A-POLD3*-pDONR221.

---

**Figure 6.  The FeS cluster influences the balance between polymerase and exonuclease activities.**
**(A)** Model of human POLD1 structure. N terminus in light grey, exonuclease domain in green, interdomain linker region in yellow, polymerase domain in blue, and C terminus in dark grey. The approximate locations of CysA, CysB, and the interaction sites with PCNA and POLD2 are highlighted in the structure. **(B)** Graphical representation of primer extension assays in the presence of increasing concentrations of dNTPs. In blue: % of primers extended; in grey: % of primers unextended; and in green: % of primers degraded.

POLD1 variants were generated by site-directed mutagenesis. The different POLD1 and CTD constructs were then cloned into pFASTbac1-based plasmids for bacmid production and for the expression of N-terminally Flag- or GST-tagged constructs in Sf9 insect cells. POLD2-T2A-POLD3-T2A-POLD4 and POLD2-T2A-POLD3 were cloned into pFASTbac1-based vectors that allowed for expression as non-tagged proteins in Sf9 insect cells. PCNA was inserted in pFASTbac1-based vectors and was expressed as an untagged or N-terminally Flag-tagged protein in Sf9 insect cells. For PCR and sequencing primers, see supplementary information (Tables S3 and S4).

### Iron incorporation assay

To assess the ability of a given protein variant of interest to bind an FeS cluster, a radioactive iron incorporation assay in Sf9 insect cells was used, similarly to what has been used previously in yeast (Pierik et al, 2009). Typically, a 20-ml culture of insect cells ($1 \times 10^6$ cells/ml) was infected with baculoviruses coding for Flag-POLD1 or GST-CTD variants in the presence of 20 $\mu$l of $^{55}$FeCl$_3$ (1 mCi/ml). The cells were collected 48 h postinfection by centrifugation, washed once with 5 ml of citrate buffer (50 mM sodium citrate and 5 mM EDTA in 1× PBS (pH 7.0)) and once with 10 ml of PBS before being resuspended in 1 ml of ice-cold lysis buffer (50 mM Tris–HCl (pH 8.0), 200 mM NaCl, 1 mM TCEP, 1 mM EDTA, 10% glycerol, 0.25% NP-40, and protease inhibitor cocktail), and incubated for 30 min on ice. After centrifugation for 30 min at 17,200 g and 4°C, radioactively labelled proteins were captured with Flag-M2 agarose beads (Sigma-Aldrich) or glutathione sepharose 4B resin (GE healthcare) for 2 h at 4°C. Subsequently, the beads were collected by centrifugation for 3 min at 250 g and 4°C, and the remaining lysate was removed completely. Beads were washed four times with 1 ml of ice-cold lysis buffer. 90% of the beads were resuspended in 1 ml of Ultima Gold scintillation liquid (PerkinElmer) and subjected to radioactivity measurement using a Tri-Carb scintillation counter (Packard). The remaining beads were analysed by SDS–PAGE to check for equal expression.

### Protein purification

Sf9 insect cells were coinfected with recombinant baculoviruses for the expression of N-terminally Flag-tagged POLD1 variants, untagged POLD2-T2A-POLD3-T2A-POLD4 or POLD2-T2A-POLD3, and untagged PCNA, where applicable. The cells were collected 48 h postinfection and snap-frozen. The cells were thawed in 1 ml of ice-cold lysis buffer (50 mM Tris–HCl (pH 8.0), 200 mM NaCl, 1 mM TCEP, 1 mM EDTA, 10% glycerol, 0.25% NP-40, and protease inhibitor cocktail). Once thawed, the cells were resuspended and incubated on ice for 30 min to allow efficient lysis. The lysed cells were subjected to 30 min of centrifugation at 17,200 g and 4°C to obtain the protein extract. Subsequently, recombinant Pol δ was captured by immunoprecipitation with Flag-M2 agarose beads (Sigma-Aldrich) for 2 h at 4°C before beads were gently spun down for 3 min at 250 g and 4°C, the remaining lysate was removed, and the beads were washed five times with 1 ml of ice-cold lysis buffer. Recombinant Pol δ was eluted with 200 $\mu$l of lysis buffer supplemented with 200 ng/$\mu$l 3× Flag peptide for 30 min at 4°C and stored

at −80°C. Purity and integrity of the purified complexes were analysed by SDS–PAGE and stained with InstantBlue protein stain (Expedeon).

N-terminally Flag-tagged PCNA was expressed and purified according to the same protocol.

### Size-exclusion chromatography

Flag-purified Pol δ and Pol δ-CS were loaded on a Superdex 200 10/300 GL size-exclusion chromatography column connected to an AKTA pure fast protein liquid chromatography system (GE Healthcare), which had been calibrated with S200 buffer (25 mM Tris–HCl (pH 8.0), 300 mM NaCl, 10% glycerol, 0.01% NP-40, 0.5 mM TCEP, and 0.5 mM EDTA). 300-$\mu$l fractions were collected and analysed by SDS–PAGE stained with InstantBlue protein stain.

### Preparation of synthetic DNA substrates and probes

HPLC-grade oligonucleotides used to prepare synthetic DNA substrates and probes are listed in the supplementary information (Table S5). Fluorescent primers had a fluorescein amidite label at the 5′-end. Primer–template substrates and dsDNA probes were annealed by heating equimolar amounts of DNA oligomers in 10 mM Tris–HCl (pH 7.5), 50 mM KCl, and 0.5 mM EDTA at 95°C for 5 min followed by slow cooling to room temperature. Annealed substrates were stored at −20°C.

### Primer extension and degradation assays

Primer extension and degradation assays were performed using fluorescently labelled primer–template substrates (Table S5). Reactions were carried out at 37°C in a 20-$\mu$l reaction volume containing 20 mM Tris–HCl (pH 8.0), 20 mM KCl, 8 mM MgCl$_2$, 0.5 mM TCEP, 0.1 mg/ml BSA, 20 nM DNA primer–template, and 2 nM of Pol δ, purified with or without PCNA. Primer extension reactions were initiated by addition of Pol δ and rapid mixing. In vitro DNA synthesis was monitored in the presence of 100 $\mu$M dNTPs over time (1, 3, 5, and 10 min). In contrast, primer degradations were monitored in the absence of dNTPs over time (1, 3, 9, and 20 min). Reactions were quenched with 20 $\mu$l of stop buffer containing 95% formamide, 0.25% bromophenol blue, and 200 nM of the single-stranded DNA competitor T61 (Table S6). Products of the enzymatic reactions were boiled and resolved on a 12% (vol/vol) DNA sequencing polyacrylamide gel (19:1 acrylamide to bis-acrylamide ratio) supplemented with 7 M urea. The gels were imaged with a Fuji FLA-9500 imager and quantified using Image Quant TL 8.0 software (GE Healthcare).

### dNTP concentration-dependent primer degradation-to-extension switch assay

Assays measuring the effect of dNTP concentration on the switching of human Pol δ between the exonuclease and polymerase active sites were performed using fluorescently labelled primer–template substrates (Table S5). Before monitoring the enzymatic activities, 40 nM of Pol δ–PCNA complexes were pre-bound to the DNA substrate at 37°C for 10 min in binding buffer containing 20 mM Tris–HCl (pH

8.0), 20 mM KCl, 0.5 mM TCEP, 0.1 mg/ml BSA, and 40 nM DNA primer–template. Subsequently, enzymatic reactions were carried out at 37°C for 5 min in a 20-$\mu$l reaction volume containing 30 mM Tris–HCl (pH 8.0), 30 mM KCl, 8 mM MgCl$_2$, 0.7 mM TCEP, 0.15 mg/ml BSA, 20 nM DNA primer–template, and 20 nM of Pol $\delta$–PCNA complexes. The reactions were performed in the presence of increasing amounts of dNTPs (0.01, 0.02, 0.05, 0.1, 0.2, 0.5, 1, 2, 5, 10, 20, 30, 40, 50, and 100 $\mu$M). Reactions were quenched with 20 $\mu$l of stop buffer containing 95% formamide, 0.25% bromophenol blue, and 200 nM of the single-stranded DNA competitor T61. Reaction products were boiled, resolved, imaged, and quantified as described above for the primer extension and degradation assays.

### Thermal inactivation assay

Complex stability was studied using a time-resolved thermal inactivation assay coupled to a single time-point primer extension reaction. DNA polymerase inactivation was performed in a 50-$\mu$l reaction volume containing 20 mM Tris–HCl (pH 8.0), 20 mM KCl, 0.5 mM TCEP, and 20 nM of Pol $\delta$–PCNA complexes. Reactions were incubated at 55°C for various amounts of time (1, 2, 3, 5, 10, 15, 20, and 25 min). Thermally inactivated Pol $\delta$–PCNA complexes were then incubated for 10 min at 25°C to cool down. The remaining enzymatic activity was monitored using single time-point primer extension assays performed for 10 min at 37°C in a 20 $\mu$l reaction volume containing 20 mM Tris–HCl (pH 8.0), 20 mM KCl, 8 mM MgCl$_2$, 0.5 mM TCEP, 0.1 mg/ml BSA, 20 nM DNA primer–template, 100 $\mu$M dNTPs, and 2 nM thermally inactivated Pol $\delta$–PCNA complexes. Reactions were quenched with 20 $\mu$l of stop buffer containing 95% formamide, 0.25% bromophenol blue, and 200 nM of the T61 competitor. Reaction products were boiled, resolved, imaged, and quantified as described above for the primer extension and degradation assays.

### Modelling of the 3D structure of POLD1

To envision the potential orientation of the CysA and CysB motifs in the C terminus of POLD1, a three-dimensional structure of the subunit was modelled using Protein Homology/AnalogY Recognition Engine V2.0 (Phyre2) (Kelley & Sternberg, 2009). In the first step, the structure of the region covering the NTD, the 3′–5′ exonuclease domain and the DNA polymerase domain (residues 77–984) was predicted with an overall confidence index of 100% based on 52% of identity with the Pol3 subunit of yeast DNA polymerase $\delta$ (PDB: 3IAY) (Swan et al, 2009). Second, the structure of the CTD fragment (935–1,096) was predicted with an overall confidence index of 100% based on 27% of identity with the Pol1 subunit of human DNA polymerase alpha (PDB: 5EXR) (Baranovskiy et al, 2016). Finally, a model of POLD1—covering residues 77–1,096—was assembled using UCSF Chimera software (Pettersen et al, 2004).

### EMSAs

EMSA reactions were performed in a 10-$\mu$l reaction volume containing 20 mM Tris–HCl (pH 8.0), 20 mM KCl, 1 mM EDTA, 0.5 mM TCEP, 20 nM 5′-fluorescein labelled DNA probes, and various amounts of Pol $\delta$, purified with or without PCNA (10, 20, 40, and 80 nM).

Reactions were incubated at 25°C for 30 min. A dye-free loading buffer containing 20 mM Tris–HCl (pH 8.0), 20 mM KCl, 1 mM EDTA, and 15% (vol/vol) Glycerol was added, and the samples were resolved on 7% (vol/vol) native polyacrylamide gels (37.5:1 acrylamide to bis-acrylamide ratio) in 0.5× TBE buffer (50 mM Tris, 50 mM Boric acid, and 0.5 mM EDTA). Gels were imaged with Fuji Film FLA-9500 imager and quantified using Image Quant TL 8.0 software (GE Healthcare).

### pSJ4-lacZα forward mutation assay

Replication fidelity was measured using a plasmid-based pSJ4-*lacZα* forward mutation assay using a modified version of the previously described pSJ3 plasmid (Keith et al, 2013). The pSJ4-*lacZα* substrate contains only a 64-nucleotide-long gap (versus a 163-nucleotide-long gap in pSJ3), which makes it very useful in measuring the fidelity of distributive DNA polymerases (Guilliam et al, 2015). The pSJ4 plasmid was gapped using NtBpu10I (Thermo Fisher Scientific) and pSJ4 competitor DNA (Table S6) following a protocol described earlier (Keith et al, 2013). Plasmid gap-filling reactions were carried out at 37°C for 30 min in a 10-$\mu$l reaction volume containing 20 mM Tris–HCl (pH 8.0), 20 mM KCl, 8 mM MgCl$_2$, 0.5 mM TCEP, 0.1 mg/ml BSA, 20 fmol of gapped pSJ4 plasmid, 100 $\mu$M of each dNTP, and 25 nM of Pol $\delta$. Plasmid gap-filling reactions with Pol $\delta$–PCNA complexes were supplemented with 500 $\mu$M ATP and 200 nM of yeast RFC (kind gift from Dr Petr Cejka [Levikova & Cejka, 2015]). Completion of the gap-filling reactions and dependency on the presence of RFC was confirmed using an analytical digestion with EcoRI (New England Biolabs) followed by 1% agarose electrophoresis. All subsequent steps of the pSJ4-*lacZα* forward mutation assay were performed as described earlier (Keith et al, 2013).

# Supplementary Information

# Acknowledgements

We thank the whole Gari lab and Stefano Ferrari for helpful discussions and critical reading of the manuscript. This project has received funding from the European Union's Horizon 2020 research and innovation programme under the Marie Skłodowska-Curie Grant Agreement No. 707299, the Swiss National Science Foundation (PP00P3_144784/1 and PP00P3_172959/1), and the Human Frontier Science Program (CDA00043/2013-C).

## Author Contributions

SK Jozwiakowski: conceptualization, resources, data curation, formal analysis, funding acquisition, validation, investigation, visualization, methodology, and writing—original draft, review, and editing.
S Kummer: resources and investigation.

K Gari: conceptualization, resources, data curation, formal analysis, supervision, funding acquisition, visualization, project administration, and writing—original draft, review, and editing.

## Conflict of Interest Statement

The authors declare that they have no conflict of interest.

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
