## [Reviewer comments · Life Science Alliance]

Life Science Alliance

Human DNA polymerase delta requires an iron-sulfur cluster for high-fidelity DNA synthesis

Stanislaw Jozwiakowski, Sandra Kummer, and Kerstin Gari

DOI: <https://doi.org/10.26508/lsa.201900321>

Corresponding author(s): Kerstin Gari, University of Zurich

Review Timeline:	Submission Date:	2019-01-28
	Editorial Decision:	2019-03-04
	Revision Received:	2019-05-07
	Editorial Decision:	2019-06-19
	Revision Received:	2019-06-26
	Accepted:	2019-06-27

Scientific Editor: Andrea Leibfried

Transaction Report:

March 4, 2019

Re: Life Science Alliance manuscript #LSA-2019-00321-T

Prof. Kerstin Gari
University of Zurich
Institute of Molecular Cancer Research
Winterthurerstrasse 190
Zurich, Zurich 8057
Switzerland

Dear Dr. Gari,

Thank you for submitting your manuscript entitled "Human DNA polymerase delta requires an iron-sulfur cluster for high-fidelity DNA synthesis" to Life Science Alliance. The manuscript was assessed by expert reviewers, whose comments are appended to this letter.

As you will see, your work overall received split opinions from the reviewers, though some concerns are shared among all three of them. While the concerns raised in the reports of reviewer #2 and #3 can get addressed in a straightforward manner, reviewer #1 does not support publication and notes issues with the methodological approaches chosen as well as inconsistencies in the dataset. Given that reviewer #1 criticizes the approaches chosen and that addressing this reviewer's concerns would require a lot of effort, I sought additional advice from reviewer #2 and #3 on these concerns. The reviewers largely agreed with reviewer #1's criticisms, but based on the advice received, I would like to invite you to provide a revised version of your work addressing the following:

- concerns of reviewer #2 and #3 should all get addressed
- please provide a SDS-PAGE gel showing presence of all four subunits of Pol delta to address rev#1, point 1 and explain the presence of the 27kDa band (rev#2); also, please provide gel-filtration evidence to show reconstitution of the four-subunit assembly as a proof-of-principle to address rev#1, point 1
- please perform a gel filtration experiment of the CS variant to provide insight into potential conformational destabilization and consequent loss of polymerase activity and discuss possible consequences in the manuscript text to address rev#1, point 2 (first half); provide control to exclude presence of contaminating polymerases (example assay where pol delta activity is restored by separate addition of purified PCNA or co-express catalytically dead version of Pol delta with PCNA) to address rev#1, point 2 (second half) and rev#1, point 3
- discuss point 4 of rev#1 and comment on presence of the two super-shifted bands in Figure 4
- discuss potential explanations for point 5 of rev#1
- respond to point 6 and 7 of rev#1 in your point-by-point response and in the discussion of your manuscript

The typical timeframe for revisions is three months, but I'd be happy to extend the revision time should this be helpful. Please note that papers are generally considered through only one revision cycle, so strong support from the referees on the revised version is needed for acceptance.

Thank you for this interesting contribution to Life Science Alliance. We are looking forward to receiving your revised manuscript.

Sincerely,

B. MANUSCRIPT ORGANIZATION AND FORMATTING:

Full guidelines are available on our Instructions for Authors page, <http://www.life-science->

Reviewer #1 (Comments to the Authors (Required)):

It was earlier shown by Baranovskiy et al that human Pol Delta has an FeS cluster in the C-terminus of the catalytic subunit, POLD1. The authors of this manuscript confirm that finding and they expand on the characterization of a human Pol delta complex lacking an FeS cluster. They claim that loss of the FeS cluster results in impaired dsDNA binding and suggest that this leads to impaired DNA synthesis. When co-expressing mutant human Pol delta with PCNA and purifying the Pol delta- PCNA complex they claim that the ability to synthesize DNA is partially rescued. Finally they suggest that the FeS cluster is important for the proofreading activity of Pol D, based on that lack of FeS cluster results in low-fidelity synthesis.

I disagree with many of the conclusions drawn by the authors of this manuscript and there are technical issues when reading the materials and methods. In summary, the conclusions drawn are at large not supported by the experimental data.

Major Concerns:

1. Figure 2a (discussed on page 4) raises some fundamental questions. It is described in the methods section that the observed proteins in the SDS-PAGE was only subject to a single step affinity purification, utilizing a FLAG-tag on the catalytic subunit. The samples are not analyzed by size-exclusion chromatography to show that a stoichiometric complex is formed. This is a standard procedure and would also improve the purity since many contaminating proteins are observed on the SDS-PAGE. I am in particular interested in whether it is known which protein is observed at around 27 kDa size? The authors claim that they have purified 4-subunit Pol delta in all eight lanes, but I can only see the smallest subunit (POLD4) in two of the lanes.

2. Same paragraph, page 4, the authors write: " To test if the FeS cluster binding variants of human Pol δ could still be structurally destabilized at higher temperatures, we subjected wild-type and variant Pol δ complexes to time-resolved thermal inactivation at 55{degree sign}C, and subsequently measured their ability to perform DNA synthesis using primer extension assays (Fig. 2b). Since the CS variant complex in the absence of PCNA had very inefficient DNA polymerase activity in a primer extension assay, we had to limit our analysis to the wild-type and FeS clusterbinding variants of Pol δ purified with PCNA.

Why exploring the enzymatic assay that could be affected by many different changes in the property of the enzyme, not only subunit interactions? Here it would be better to heat-treat the

protein complexes and then pass it over a size-exclusion column to ask whether any of the subunits dissociate. This is a very relevant experiment as it was previously shown that the FeS cluster in yeast Pol delta is important for the interaction between the catalytic subunit and the accessory subunits. Based on the yeast results, subunits in human Pol delta may dissociate at an elevated temperature, in particular if the FeS cluster is destabilized.

Is the lack of DNA polymerase activity in the CS-variant complex due to protein misfolding? Again a gel-filtration experiment would bring clarity. Minor point, the phrase "To test if the FeS cluster binding variants of human Pol δ could" suggests that the variants are binding FeS cluster and the opposite is shown in Fig 1c.

It is worrying that the Pol delta CS-variant by itself has a weak polymerase activity, but when purified in a complex with PCNA the mutant has good activity. Are the authors certain that the Pol delta -PCNA complex fractions are devoid of any contamination from other DNA polymerases from the insect cells? PCNA is easy to express in E. coli and can thus be purified from a source where no additional PCNA binding proteins will contaminate the protein prep. Thus, a good biochemical practice is to use Pol Delta from one protein prep and then compare the activity with and without the addition of PCNA. By this method the risk for contamination from other DNA polymerases are eliminated since, if needed, the PCNA prep can be tested for DNA Polymerase activity prior to addition to the assays with Pol Delta.

I strongly disagree with the approach taken by the authors to compare a protein prep with only Pol Delta with another protein prep where Pol Delta is purified with PCNA. This is strengthened by the observed differences in purity shown in Fig 2a.

3. Figure 3 b and c, Again, the experiment should have been made with the same Pol Delta preps, with or without adding PCNA that was purified separately.

4. First paragraph page 5, Figure 4 a and b, The relevant substrate for Pol Delta when binding DNA is the primed substrate with a 3'-OH end and a protruding single-stranded DNA template. The authors conclude "We first tested binding of the Pol δ variants to a primer-template substrate in the absence or presence of PCNA (Fig. 4a,b), but did not observe any obvious differences between the variants." Case closed! The primer-template substrate is the natural substrate during DNA replication or any other instance when Pol Delta synthesizes DNA and it has duplex DNA which Pol Delta holds on to with the help of the thumb domain. What biological relevance the observed difference in binding to duplex DNA has is unclear to me and I disagree with the author's interpretation. Furthermore, the EMSAs in Fig 4 a and b show two different migrating bands, suggesting that two different complexes are formed with the DNA as the concentration of protein increases. What are these complexes? Why is there two different bands?

5. Figure 5, please show a SDS-PAGE with the purified three-subunit Pol delta.

It is striking that the HY variant, even when only having a 20% loss in Fe binding (fig 1c) show a defect in exonuclease activity that is comparable to the CS variant that has a 90% loss of Fe binding (Fig 1c). Would not this observation rather suggest that the deficiency to exonuclease activity is unrelated to the FeS cluster.

6. The authors perform an in vitro gap-filling assay to score replication errors made by the studied DNA polymerase. In this case the authors compare the Pol Delta variants that are purified in complex with PCNA. The gap-filling assay is outlined in Fig 6a and there it can be seen that the gap is present in a circular plasmid. Thus, in order for PCNA to be loaded must the clamp-loader, RFC, be added to the assay. Under normal circumstances RFC loads free PCNA and Pol Delta binds

thereafter to PCNA on the DNA. Here the authors have chosen to add yeast RFC. However, yeast RFC is unable to load human PCNA onto the DNA. Furthermore, I am uncertain whether human RFC could load PCNA while bound to Pol Delta because they both interact with the same side of PCNA (competition). To summarize, the gap-filling will be distributive since Pol delta is not very processive without the interaction with PCNA bound to DNA as can be read in the papers that are referred to by the authors. Again the authors chose to only analyze the fidelity of the Pol Delta-PCNA complex, with or without variants. And again, I am very concerned that there may have been contaminating DNA polymerases in the assay and this would certainly influence the distribution of replication errors in the assay.

7. The last paragraph in the results section, figure 8, and the discussion: The proposal that the FeS cluster is potentially required for the controlled switching between DNA exonuclease and polymerase active sites is an intriguing idea, but should be removed since there is no support for that model in the presented results. Making amino acid substitutions that stop FeS cluster from binding is very different from changing a redox potential. To open this discussion, the authors should first demonstrate that a changed redox potential results in a shift from polymerase to exonuclease activity or the opposite.

Reviewer #2 (Comments to the Authors (Required)):

The paper by Jozwiakowski and Gari reports a biochemical analysis of the FeS cluster of human Pol delta. FeS clusters have been found to be frequently present in DNA repair and replication proteins, and all B family replicative DNA polymerases, as well as DNA primase, contain one. The role of the FeS cluster in polymerase/primase activity is an area of intense research, centred principally on whether the FeS clusters have structural roles or are also redox-active, with their oxidation states possibly regulating polymerase activity.

The authors show evidence that POLD1, the large catalytic subunit of the multi-subunit Pol delta enzyme, contains a FeS cluster, thus supporting previous published evidence (Netz, 2011), and that a single-point mutation, derived from the yeast Pol delta, that abolishes FeS cluster incorporation, reduces greatly the thermal stability of the enzyme, as shown by loss of polymerase activity. They further show that impairment of the FeS cluster by mutations that either abolish or reduce FeS cluster incorporation cause impaired double-stranded DNA binding, reduced DNA synthesis, and skewed activity mode, favouring DNA synthesis over proof-reading.

The authors present a nice series of well-executed experiments, with data that support the authors' conclusions. Altogether, it represents a significant contribution to the field of replicative polymerases, that is still afflicted by controversies surrounding the topic of the putative redox role of the FeS clusters. I don't have any major criticism, but I do have some points that I would like the authors to address, to improve the readability of their manuscript.

POINTS TO ADDRESS

It is better to avoid referring to Pol delta as a 'holo-enzyme', which can mean different things in different contexts. Please refer to Pol delta as a multi-subunit polymerase, or use the term 'quaternary structure'. Equally, avoid referring to Pol delta generically as a 'complex'.

In the Introduction, the sentence 'defective in UV-induced mutagenesis' should actually read 'defective in repair of UV-induced mutagenesis'. Please clarify.

In figure 2a, what is the origin of the band just below PCNA, but in the right-hand half of the gel (samples without PCNA). Please annotate the gel.

The authors should clarify to themselves and to the reader what the likely consequence of the HY/HW mutations is. In the text they refer variously to the HY/HW mutants as having an 'aberrant FeS cluster' or a cluster 'with incorrect coordination'. The most likely consequence of the HY/HW mutations is a local conformational distortion in the FeS cluster binding motif, leading to altered cysteine ligand geometry and consequently reduced levels of FeS incorporation. So rather than referring to aberrant/incorrect FeS clusters, they should refer to aberrant or altered FeS cluster binding motifs.

A similar confusion generated by inaccurate language arises in the experiments that measure DNA synthesis at high temperature (55 degrees). Here the authors refer to the 'thermal stability' of the enzyme. However, this is not what they measured, which is levels of DNA synthesis. Given that the CS mutant activity in DNA synthesis has greatly reduced resistance to temperature, the authors infer lower thermal stability. In reality, the loss of the FeS cluster is likely to cause only modest and local impairment in the structural stability of this large multi-subunit enzyme. So if the authors want to be able to refer to the thermal stability of Pol delta, they should measure it. Why the local unfolding of the FeS cluster binding motif should have such a drastic negative effect on DNA synthesis is an interesting question! The interesting effect of PCNA in rescuing the activity of the CS mutant, seen in various assays, is probably due a compensatory, stabilising effect of PCNA binding to the neighbouring PIP box.

In describing the experiments of Figure 3, it is inaccurate to say that primers were extended by the CS mutant to N+2 or digested to N-4, all products in between were also detected, as expected. Please correct by saying 'extended up to' two nucleotides and 'degraded up to' four nucleotides. In the same experiment, it is not clear why this effect should be caused by the enzyme being less stably bound to DNA.

The decision to move some of the data in the supplementary information seems arbitrary and not helpful to the reader. For instance, why did the authors decide to move the ssDNA EMSA to the supplementary (Suppl fig 2a, b)? Please move to main figure 4, so that all EMSA experiments can be seen side by side.

Equally, it is not clear why the authors present the exonuclease data of the three-subunit Pol delta in Fig. 5, but the same data for the four-subunit enzyme in Supplementary figure 3. Please present all the data in Figure 5.

The 3D model of Pol delta that the authors propose at the end provides very limited insight, because based on the structure of Pol alpha within Pol alpha/primase, which is probably in a very different conformation. What I find intriguing about the data is the fact that mutations in the FeS cluster binding motif, located in the CTD, affect the catalytic functions of nucleotide polymerisation/proofreading, which are contained within the polymerase domain. In the model CTD and polymerase domain are far apart. It is difficult to envisage at present how this can happen without large scale conformational rearrangements. The authors don't comment on this, but I would invite them to do so.

I agree with the authors that the model postulating a role of the FeS cluster in translesion synthesis in the presence of UV damage is intriguing, Have the authors considered performing an experiment

where they look at the ability of the HY/HW mutants to perform translesion synthesis after UV damage? Based on their observations, the mutants should be better able to cope with such a damaged template than the wild-type polymerase. This is not required for publication but I would invite the authors to perform such an experiment, or to include such data if they already have it.

I was also wondering if the authors has measured the affinity of the wild-type and mutant proteins for PCNA. Even a simple pull-down experiment would be informative of whether the CS, HY, HW mutations have impacted the ability to bind PCNA.

Reviewer #3 (Comments to the Authors (Required)):

Jozwiakowski and Gari investigate the role of DNA Pol delta's FeS cluster. The subject of the study is timely and important to the understanding of the functions of mammalian replicases. However, there are problems with the paper, including inconsistent results, incomplete methods description, and overstatements. These points should be addressed before publication.

See specific comments below.

1. Page 3, the authors state: "... other residues within the FeS cluster binding pocket can potentially stabilise the co-factor, e.g. histidine residue through hydrogen bonding (Bak & Elliott, 2013)". However, the paper the authors refer to (Bak & Elliott, 2013), indicates the opposite: "The mitoNEET [2Fe-2S] cluster demonstrates proton-coupled electron transfer (PCET) and marked cluster instability, which have both been linked to the single His ligand."

2. Data in Fig. 1c and Supplementary Fig. 1b indicate that there is a low, although measurable amount of iron bound to the C1076S. Accordingly, without measuring the background level of radioactivity the authors should not claim a complete loss of the Fe-S cluster in C1076S pol d.

3. Page 7, "...we hypothesize that the FeS cluster could potentially be required to control the conformational flexibility of the linker and consequently regulate the switch between DNA polymerization and proofreading." In order to test this hypothesis,..." The conducted experiment (Fig. 7b, Supplementary Fig. 5) does not provide any information about the conformation of the flexible linker, or its role in the optimal positioning of the enzyme on the DNA substrate (another presumption of the authors) and in regulating of the switch between polymerase and exonuclease.

4. It appears that there is more unreacted substrate in reactions with 20nM CS Pol d + PCNA and 100 μ M dNTPs (supplementary Fig. 5b), than with 10-fold less enzyme (2 nM CS + PCNA and 100 μ M dNTPs) at 5 min time point (Fig. 3c). Why is that? These results seem to be inconsistent.

5. The fidelity measurements are difficult to evaluate due to the incomplete description of the treatment of the data, as well as the data presentation. The mutant frequencies for the three independent determinations should be presented to allow evaluation of the reproducibility of the measurements. Despite the fact that the method was previously published, it would be helpful if the authors

included the formula for calculating the error rate in the legend to Table 1. Also, the value of the subtracted background mutation frequency for the pSJ4 plasmid-based substrate should be stated. Was the assay based on the pSJ4 plasmid published previously? If so, please add reference. If not, a more detailed description should be included.

It is not clear how the target size (145) used for calculating the error rates was determined. The authors state that the target size-the number of detectible sites- is for all types of possible mutations (base substitutions, deletions and insertions). However, supplementary Table 2 lists "mixed" under "mutation types". There is no explanation what errors are considered to be in the "mixed". It is not clear how the target size is determined for the "mixed" category.

Minor points

5. Page 5, "To investigate the influence of the Fe-S cluster on Pol d's proofreading abilities, we purified...." An enzyme's exonuclease activity is not the same as its proofreading ability.

6. Legend to supplementary Fig. 4 should contain explanation of the annotations of the mutational spectra. (blue)

7. The model in Fig. 7a and its description in the text would be easier to follow if the polymerase subdomains (palm, fingers, thumb) were color coded (e.g. different shades of blue).

Reviewer #1 (Comments to the Authors (Required)):

It was earlier shown by Baranovskiy et al that human Pol Delta has an FeS cluster in the C-terminus of the catalytic subunit, POLD1. The authors of this manuscript confirm that finding and they expand on the characterization of a human Pol delta complex lacking an FeS cluster. They claim that loss of the FeS cluster results in impaired dsDNA binding and suggest that this leads to impaired DNA synthesis. When co-expressing mutant human Pol delta with PCNA and purifying the Pol delta- PCNA complex they claim that the ability to synthesize DNA is partially rescued. Finally they suggest that the FeS cluster is important for the proofreading activity of Pol D, based on that lack of FeS cluster results in low-fidelity synthesis.

I disagree with many of the conclusions drawn by the authors of this manuscript and there are technical issues when reading the materials and methods. In summary, the conclusions drawn are at large not supported by the experimental data.

Major Concerns:

1. Figure 2a (discussed on page 4) raises some fundamental questions. It is described in the methods section that the observed proteins in the SDS-PAGE was only subject to a single step affinity purification, utilizing a FLAG-tag on the catalytic subunit. The samples are not analyzed by size-exclusion chromatography to show that a stoichiometric complex is formed. This is a standard procedure and would also improve the purity since many contaminating proteins are observed on the SDS-PAGE. I am in particular interested in whether it is known which protein is observed at around 27 kDa size? The authors claim that they have purified 4-subunit Pol delta in all eight lanes, but I can only see the smallest subunit (POLD4) in two of the lanes.

We had initially purified the Pol delta variants with a two-step purification: Flag-PD and S200 gel filtration, as seen in **attachment 1** for the wild-type and CS variant enzyme (purified in the presence of PCNA). While the complexes were stable and eluted stoichiometrically in a defined peak, their purity did not appear to be improved, as compared to a one-step Flag-pull-down purification. Given that the additional step had prolonged the procedure (which we feared may affect the stability of the FeS cluster) and led to a dilution of the samples without further purification, we had decided to abstain from a second purification step.

We agree with the reviewer that the SDS-PAGE presented did not show sufficiently well the presence of POLD4, which is at least in part due to the weak staining of POLD4 due to its small size. Therefore, we now provide a western blot for POLD4 (**Figure 2A, bottom**), which clearly shows the presence of POLD4 in all samples, although less POLD4 appears to be present in the CS variant sample without PCNA. In the results section, we point this out and suggest that the CS variant complex may be partially destabilised in the absence of PCNA.

Like the reviewer we were puzzled by the additional band at around 27 kDa that is enriched in samples without PCNA. Initially we supposed that the band arises from proteolysis of Pol delta and that the proteolysis is somehow more pronounced in the samples without PCNA.

To identify the protein, we ran the wild-type Pol delta sample (without PCNA) on an SDS-PAGE, cut the band out and sent it for mass spectrometry analysis. Apart from two human POLD2 peptides, indicative of protein degradation of this subunit, and two peptides of an unidentified *Bacillus cereus* protein, the analysis picked up eight peptides from baculoviral PCNA (**attachment 2**). While we were initially surprised to find that baculoviral PCNA is closely enough related to human Pol delta to bind to it, on second thoughts this would readily explain why we see less of it in the samples that contain human PCNA. We now mark the corresponding band with an asterisk in the SDS-PAGE and add this information to the figure legend.

2. Same paragraph, page 4, the authors write: " To test if the FeS cluster binding variants of human Pol δ could still be structurally destabilized at higher temperatures, we subjected wild-type and variant Pol δ complexes to time-resolved thermal inactivation at 55{degree sign}C, and subsequently measured their ability to perform DNA synthesis using primer extension assays (Fig. 2b). Since the CS variant complex in the absence of PCNA had very inefficient DNA polymerase activity in a primer extension assay, we had to limit our analysis to the wild-type and FeS clusterbinding variants of Pol δ purified with PCNA.

Why exploring the enzymatic assay that could be affected by many different changes in the property of the enzyme, not only subunit interactions? Here it would be better to heat-treat the protein complexes and then pass it over a size-exclusion column to ask whether any of the subunits dissociate. This is a very relevant experiment as it was previously shown that the FeS cluster in yeast Pol delta is important for the interaction between the catalytic subunit and the accessory subunits. Based on the yeast results, subunits in human Pol delta may dissociate at an elevated temperature, in particular if the FeS cluster is destabilized.

We have performed this experiment (**attachment 3**). However, the resulting size exclusion chromatography showed that subjecting human Pol delta preparations (both wild-type and CS variant) to 3 minutes of heat stress at 55°C caused an aggregation of the proteins. This was manifested by a shift in the elution profile to earlier fractions (void volume), while the fully assembled hetero-tetrameric polymerase normally eluted in fractions 33-38 (see also **attachment 1**). Additionally, a lower signal for all fractions was observed on the SDS gel. This may suggest that a significant portion of the proteins was retained on the filter of the size exclusion column as insoluble aggregates. Indeed, in the inverted flow wash following the experiment we observed a strong signal at an absorbance of 280 nm (quite likely insoluble aggregates of human Pol delta).

Is the lack of DNA polymerase activity in the CS-variant complex due to protein misfolding? Again a gel-filtration experiment would bring clarity.

We think it is unlikely that Pol delta-CS is globally misfolded for two reasons: First, it behaves similarly as the wild-type enzyme on a size exclusion column (**attachment 1**); and second, our new data indicate that polymerase activity can be restored by the addition of ectopically purified PCNA (**Figure S2**). Interestingly, this experiment indicates that Pol delta-

CS is almost completely dependent on the presence of PCNA, whereas the wild-type enzyme shows significant DNA polymerase activity when tested without PCNA.

Minor point, the phrase " To test if the FeS cluster binding variants of human Pol δ could" suggests that the variants are binding FeS cluster and the opposite is shown in Fig 1c.

We agree with the reviewer that this sentence was misleading. We have changed our terminology throughout the manuscript and now refer to either "CysB variants" or variants with "alterations in the FeS cluster-binding pocket/motif".

It is worrying that the Pol delta CS-variant by itself has a weak polymerase activity, but when purified in a complex with PCNA the mutant has good activity. Are the authors certain that the Pol delta -PCNA complex fractions are devoid of any contamination from other DNA polymerases from the insect cells? PCNA is easy to express in *E. coli* and can thus be purified from a source where no additional PCNA binding proteins will contaminate the protein prep. Thus, a good biochemical practice is to use Pol Delta from one protein prep and then compare the activity with and without the addition of PCNA. By this method the risk for contamination from other DNA polymerases are eliminated since, if needed, the PCNA prep can be tested for DNA Polymerase activity prior to addition to the assays with Pol Delta.

I strongly disagree with the approach taken by the authors to compare a protein prep with only Pol Delta with another protein prep where Pol Delta is purified with PCNA. This is strengthened by the observed differences in purity shown in Fig 2a.

We have followed the reviewer's advice and purified PCNA to add it to primer extension reactions with Pol delta-CS (purified in the absence of PCNA). Since all our proteins were purified in *Sf9* insect cells and we had expression constructs available, we chose to purify human Flag-PCNA from *Sf9* insect cells, rather than *E. coli*. As can be seen on an SDS-PAGE (**Figure S2A**), the purity of the recombinant protein was close to homogeneity and the preparation free of any visible contaminating proteins. Nevertheless, to exclude the possibility that the preparation was contaminated with an *Sf9* DNA polymerase or exonuclease, we ran primer extension and primer degradation assays with purified Flag-PCNA. As can be seen in **attachment 4**, no DNA template-dependent DNA polymerase activity and no DNA nuclease activity were detected in reactions containing increasing amounts of Flag-PCNA.

When added to primer extension assays with Pol delta-CS (purified without PCNA), Flag-tagged PCNA was able to restore polymerase activity of the CS variant enzyme (**Figure S2B**). Taken together, these control experiments strongly suggest that the functional difference between the Pol delta-CS preparation purified with PCNA, as compared to the one purified without PCNA, is indeed due to the presence of PCNA.

3. Figure 3 b and c, Again, the experiment should have been made with the same Pol Delta preps, with or without adding PCNA that was purified separately.

We have done this experiment (see answer to point 2).

4. First paragraph page 5, Figure 4 a and b, The relevant substrate for Pol Delta when binding DNA is the primed substrate with a 3'-OH end and a protruding single-stranded DNA template. The authors conclude " We first tested binding of the Pol δ variants to a primer-template substrate in the absence or presence of PCNA (Fig. 4a,b), but did not observe any obvious differences between the variants." Case closed ! The primer-template substrate is the natural substrate during DNA replication or any other instance when Pol Delta synthesizes DNA and it has duplex DNA which Pol Delta holds on to with the help of the thumb domain. What biological relevance the observed difference in binding to duplex DNA has is unclear to me and I disagree with the author's interpretation. Furthermore, the EMSAs in Fig 4 a and b show two different migrating bands, suggesting that two different complexes are formed with the DNA as the concentration of protein increases. What are these complexes? Why is there two different bands?

We agree with the reviewer that the biologically relevant substrate for Pol delta is a primed DNA substrate. However, it should be stressed that Pol delta contacts a primer-template substrate both in the double-stranded region (with the thumb and C-terminal domains of POLD1) and the single-stranded region (with the N-terminal and exonuclease domains of POLD1). Our choice to study binding of Pol delta to ssDNA and dsDNA was therefore not meant to reflect on any biological relevance, but to dissect potential differences between the wild-type enzyme and the CysB variants with respect to their ability to bind to either of the two DNA binding sites. A similar approach was taken previously to study human PrimPol (Keen *et al.*, 2014).

Although the structure of the four-subunit human enzyme bound to primed DNA is not available, it is known that the catalytic subunit of yeast Pol delta alone occupies about 20 nt of dsDNA and 20 nt of ssDNA (Swan *et al.*, 2009). We therefore designed a primer-template substrate that contains a fluorescently labelled primer (42 nt) annealed to a complementary template (61 nt) such that it provides a long enough region of double-stranded DNA (42 nt) and single-stranded DNA (19 nt) for the stable binding of the four-subunit enzyme alone or associated with PCNA. With such a substrate, however, it is unlikely to detect partial defects in binding to either of the two DNA binding interfaces, since the affinity for the other interface will most likely be sufficient to confer binding to the primer-template probe. Indeed, the difference between the wild-type enzyme and Pol delta-CS in binding to the primer-template substrate was marginal, except that at higher protein concentrations two distinct shifts were clearly discernible for the wild-type enzyme, but not (or to a much lower extent) for Pol delta-CS. The most logical explanation for the higher shift seemed to be the simultaneous binding of two enzymes to the primer-template substrate, whereby one is bound at the ss/dsDNA junction and the other one most likely at the dsDNA region upstream of it. The fact that Pol delta-CS seemed less able than the wild-type enzyme to produce such a super-shift suggested to us that it may have some partial defect in binding to either dsDNA or ssDNA that does not become immediately apparent with the natural substrate. Only by using a dsDNA substrate we were able to show that Pol delta-CS has a partial DNA binding defect that can be restored by the addition of PCNA. We have changed the relevant section in the manuscript to better explain the rationale behind our approach.

5. Figure 5, please show a SDS-PAGE with the purified three-subunit Pol delta. It is striking that the HY variant, even when only having a 20 % loss in Fe binding (fig 1c) show a defect in exonuclease activity that is comparable to the CS variant that has a 90 % loss of Fe binding (Fig 1c). Would not this observation rather suggest that the deficiency to exonuclease activity is unrelated to the FeS cluster.

We now show an SDS-PAGE of the purified three-subunit enzymes (**Figure 4D**).

In our view, the fact that the HY variant has greatly reduced exonuclease activity despite being able to bind an FeS cluster suggests that the mere presence of an FeS cluster is not sufficient for proper exonuclease activity. Instead we suspect that the surrounding amino acids are important for the correct positioning of the FeS cluster within the FeS cluster-binding pocket.

While in Pol delta-CS the cofactor is most likely lost, which leads to drastic structural changes in the FeS cluster pocket, it is conceivable that in the histidine variants – possibly due to the absence of a positive charge – the FeS cluster-binding pocket undergoes smaller, but still functionally relevant, structural changes despite a bound FeS cluster. Since the FeS cluster-binding pocket is located in the flexible region that connects the C-terminal part of POLD1 with the catalytic domain of Pol delta we believe that it may influence conformational switching between DNA synthesis and proofreading, but a detailed understanding about such a function will require extensive structural studies.

6. The authors perform an *in vitro* gap-filling assay to score replication errors made by the studied DNA polymerase. In this case the authors compare the Pol Delta variants that are purified in complex with PCNA. The gap-filling assay is outlined in Fig 6a and there it can be seen that the gap is present in a circular plasmid. Thus, in order for PCNA to be loaded must the clamp-loader, RFC, be added to the assay. Under normal circumstances RFC loads free PCNA and Pol Delta binds thereafter to PCNA on the DNA. Here the authors have chosen to add yeast RFC. However, yeast RFC is unable to load human PCNA onto the DNA. Furthermore, I am uncertain whether human RFC could load PCNA while bound to Pol Delta because they both interact with the same side of PCNA (competition). To summarize, the gap-filling will be distributive since Pol delta is not very processive without the interaction with PCNA bound to DNA as can be read in the papers that are referred to by the authors. Again the authors chose to only analyze the fidelity of the Pol Delta-PCNA complex, with our without variants. And again, I am very concerned that there may have been contaminating DNA polymerases in the assay and this would certainly influence the distribution of replication errors in the assay.

Previous studies have shown experimentally that the yeast clamp loader Rfc can efficiently load human PCNA on circular primed DNA (Yoder and Burgers, 1991) and on circular DNA (Dzantiev et al, 2004).

In our plasmid-based fidelity assay, the gapped region of the plasmid contains a unique EcoRI cutting site that allows us to monitor gap filling (**attachment 5A**). In the absence of yeast Rfc, the four-subunit variants of Pol delta associated with PCNA were very inefficient in filling up the gapped region, with the exception of some DNA synthesis observed for the wild-type (exo+) enzyme and the HW (exo+) variant. In contrast, in the presence of yeast Rfc

and ATP all six analysed variants were efficient in gap filling and, hence, rendered the plasmid sensitive to EcoRI digestion (**attachment 5B**).

7. The last paragraph in the results section, figure 8, and the discussion: The proposal that the FeS cluster is potentially required for the controlled switching between DNA exonuclease and polymerase active sites is an intriguing idea, but should be removed since there is no support for that model in the presented results. Making amino acid substitutions that stop FeS cluster from binding is very different from changing a redox potential. To open this discussion, the authors should first demonstrate that a changed redox potential results in a shift from polymerase to exonuclease activity or the opposite.

We thank the reviewer for this comment, and have carefully revised the discussion. We now rather discuss the structure/function-related role of the co-factor, *i.e.* how structural changes in the FeS cluster-binding pocket may affect the catalytic function of Pol delta. In this context, we agree with the reviewer that we do not have the direct proof for a role of the FeS cluster in the switching between DNA polymerisation and proofreading. However, given the predicted positioning of the cofactor and our observation from the exonuclease-to-polymerase switching experiment (**Figure 6**), we are keen to speculate on a hypothetical relevance of the FeS cluster in this process and to point out possible future directions. In addition, we are careful not to claim a redox regulation of Pol delta *via* its FeS cluster. On the other hand, we describe a highly speculative scenario, in which the FeS cluster may get damaged and lost upon oxidative stress. Our studied variant Pol delta-CS should serve as a good model to envision the potential outcome of such an extreme scenario.

Reviewer #2 (Comments to the Authors (Required)):

The paper by Jozwiakowski and Gari reports a biochemical analysis of the FeS cluster of human Pol delta. FeS clusters have been found to be frequently present in DNA repair and replication proteins, and all B family replicative DNA polymerases, as well as DNA primase, contain one. The role of the FeS cluster in polymerase/primase activity is an area of intense research, centred principally on whether the FeS clusters have structural roles or are also redox-active, with their oxidation states possibly regulating polymerase activity.

The authors show evidence that POLD1, the large catalytic subunit of the multi-subunit Pol delta enzyme, contains a FeS cluster, thus supporting previous published evidence (Netz, 2011), and that a single-point mutation, derived from the yeast Pol delta, that abolishes FeS cluster incorporation, reduces greatly the thermal stability of the enzyme, as shown by loss of polymerase activity. They further show that impairment of the FeS cluster by mutations that either abolish or reduce FeS cluster incorporation cause impaired double-stranded DNA binding, reduced DNA synthesis, and skewed activity mode, favouring DNA synthesis over proof-reading.

The authors present a nice series of well-executed experiments, with data that support the authors' conclusions. Altogether, it represents a significant contribution to the field of

replicative polymerases, that is still afflicted by controversies surrounding the topic of the putative redox role of the FeS clusters. I don't have any major criticism, but I do have some points that I would like the authors to address, to improve the readability of their manuscript.

POINTS TO ADDRESS

It is better to avoid referring to Pol delta as a 'holo-enzyme', which can mean different things in different contexts. Please refer to Pol delta as a multi-subunit polymerase, or use the term 'quaternary structure'. Equally, avoid referring to Pol delta generically as a 'complex'.

We thank the reviewer for this comment and have changed the wording throughout the text.

In the Introduction, the sentence 'defective in UV-induced mutagenesis' should actually read 'defective in repair of UV-induced mutagenesis'. Please clarify.

In the paper by Stepchenkova and colleagues, the *pol3-13* strain is indeed shown to be defective in UV-induced mutagenesis. On a mechanistic level, the authors suggest that the FeS cluster may be required for the accurate switching from Pol delta to Pol zeta. As a consequence of a faulty switching mechanism, UV-induced (Pol zeta-dependent) mutagenesis would be lower in *pol3-13*.

In figure 2a, what is the origin of the band just below PCNA, but in the right-hand half of the gel (samples without PCNA). Please annotate the gel.

Like the reviewer we were puzzled by the additional band at around 27 kDa that is enriched in samples without PCNA. Initially we supposed that the band arises from proteolysis of Pol delta and that the proteolysis is somehow more pronounced in the samples without PCNA. To identify the protein, we ran the wild-type Pol delta sample (without PCNA) on an SDS-PAGE, cut the band out and sent it for mass spectrometry analysis. Apart from two human POLD2 peptides, indicative of protein degradation of this subunit, and two peptides of an unidentified *Bacillus cereus* protein, the analysis picked up eight peptides from baculoviral PCNA (**attachment 2**). While we were initially surprised to find that baculoviral PCNA is closely enough related to human Pol delta to bind to it, on second thoughts this would readily explain why we see less of it in the samples that contain human PCNA. We now mark the corresponding band with an asterisk in the SDS-PAGE and add this information to the figure legend.

The authors should clarify to themselves and to the reader what the likely consequence of the HY/HW mutations is. In the text they refer variously to the HY/HW mutants as having an 'aberrant FeS cluster' or a cluster 'with incorrect coordination'. The most likely consequence

of the HY/HW mutations is a local conformational distortion in the FeS cluster binding motif, leading to altered cysteine ligand geometry and consequently reduced levels of FeS incorporation. So rather than referring to aberrant/incorrect FeS clusters, they should refer to aberrant or altered FeS cluster binding motifs.

We thank the reviewer for this important comment. We have added a tentative explanation as to what the consequence of histidine replacement is (page 4): "This may suggest that replacing histidine 1066 with tyrosine or tryptophan induces structural distortions in the CysB motif that lead to altered cysteine ligand geometry and reduced FeS cluster binding." We have also changed our terminology throughout the manuscript and now refer to either "CysB variants" or variants with "alterations/structural changes in the FeS cluster-binding pocket/motif".

A similar confusion generated by inaccurate language arises in the experiments that measure DNA synthesis at high temperature (55 degrees). Here the authors refer to the 'thermal stability' of the enzyme. However, this is not what they measured, which is levels of DNA synthesis. Given that the CS mutant activity in DNA synthesis has greatly reduced resistance to temperature, the authors infer lower thermal stability. In reality, the loss of the FeS cluster is likely to cause only modest and local impairment in the structural stability of this large multi-subunit enzyme. So if the authors want to be able to refer to the thermal stability of Pol delta, they should measure it. Why the local unfolding of the FeS cluster binding motif should have such a drastic negative effect on DNA synthesis is an interesting question! The interesting effect of PCNA in rescuing the activity of the CS mutant, seen in various assays, is probably due a compensatory, stabilising effect of PCNA binding to the neighbouring PIP box.

We are grateful to the reviewer for this important comment. Indeed, multimeric enzymes subjected to heat stress typically first undergo de-oligomerisation and – only when subjected for long enough to sufficiently high temperatures – experience a complete structural denaturation. Therefore, it is possible that the lower thermal resistance observed for the CS variant is caused by a more rapid de-oligomerisation of the four-subunit structure. However, given that DNA synthesis by Pol delta-CS is highly dependent on PCNA, an alternative explanation is that Pol delta-CS is particularly affected by the dissociation of PCNA, whereas the wild-type enzyme and the histidine variants can still efficiently synthesise DNA in the absence of PCNA (see also **Figure 2C**).

We have changed the relevant paragraph of the results section (page 4/5) and abstain from using the term "thermal stability".

In describing the experiments of Figure 3, it is inaccurate to say that primers were extended by the CS mutant to N+2 or digested to N-4, all products in between were also detected, as expected. Please correct by saying 'extended up to' two nucleotides and 'degraded up to' four nucleotides. In the same experiment, it is not clear why this effect should be caused by the enzyme being less stably bound to DNA.

We have changed the description following the reviewer's advice:

"a significant portion of the primers was extended only up to two nucleotides (N+2) or degraded up to four nucleotides (N-4)"

The decision to move some of the data in the supplementary information seems arbitrary and not helpful to the reader. For instance, why did the authors decide to move the ssDNA EMSA to the supplementary (Suppl fig 2a, b)? Please move to main figure 4, so that all EMSA experiments can be seen side by side.

We have moved the EMSAs with ssDNA to the main part of the manuscript and show now EMSAs with all DNA substrates in **Figure 3**.

Equally, it is not clear why the authors present the exonuclease data of the three-subunit Pol delta in Fig. 5, but the same data for the four-subunit enzyme in Supplementary figure 3. Please present all the data in Figure 5.

We have moved the exonuclease assays with the four-subunit enzymes to **Figure 4**.

The 3D model of Pol delta that the authors propose at the end provides very limited insight, because based on the structure of Pol alpha within Pol alpha/primase, which is probably in a very different conformation. What I find intriguing about the data is the fact that mutations in the FeS cluster binding motif, located in the CTD, affect the catalytic functions of nucleotide polymerisation/proofreading, which are contained within the polymerase domain. In the model CTD and polymerase domain are far apart. It is difficult to envisage at present how this can happen without large scale conformational rearrangements. The authors don't comment on this, but I would invite them to do so.

We do not necessarily expect large conformational rearrangements upon alterations in the FeS cluster-binding motif. In our view, already small structural rearrangements in the FeS cluster-binding pocket and the flexible linker region could suffice to alter the alignment of Pol delta on the DNA substrate and, hence, affect its catalytic activities. We have revised the relevant results section (page 7) to make our rationale clearer.

I agree with the authors that the model postulating a role of the FeS cluster in translesion synthesis in the presence of UV damage is intriguing. Have the authors considered performing an experiment where they look at the ability of the HY/HW mutants to perform translesion synthesis after UV damage? Based on their observations, the mutants should be better able to cope with such a damaged template than the wild-type polymerase. This is not required for publication but I would invite the authors to perform such an experiment, or to include such data if they already have it.

We completely agree with the comment of the reviewer, but have so far not tested the TLS capability of Pol delta-HY/HW. We can therefore not tell whether the partially defective proofreading function of these variants would be sufficient to remove the kinetic barrier of proofreading, which would be necessary for Pol delta to be able to perform TLS. We anticipate that Pol delta-HY/HW could be potentially more efficient in the bypass of lesions that do not distort the DNA template extensively, such as abasic sites or 8-oxo-dG. However, in the case of bulky UV-induced lesions, the narrow cavity of the catalytic centre would be the major limitation for efficient bypass. That being said, more experimental effort will be required to address this question.

I was also wondering if the authors has measured the affinity of the wild-type and mutant proteins for PCNA. Even a simple pull-down experiment would be informative of whether the CS, HY, HW mutations have impacted the ability to bind PCNA.

We have performed a co-immunoprecipitation experiment with Flag-tagged POLD1 variants and PCNA co-expressed in *Sf9* insect cells (**attachment 6**). Unfortunately, POLD1 and PCNA on their own do not seem to associate in a sufficiently stable way with each other to detect an interaction (above background) with this approach. Perhaps more sensitive biophysical methods could provide an answer for this important question. What we can say is that in the context of the three- or four-subunit structure, PCNA seems to interact similarly with the wild-type enzyme and all three variants (**Figures 2A and 4D**).

Reviewer #3 (Comments to the Authors (Required)):

Jozwiakowski and Gari investigate the role of DNA Pol delta's FeS cluster. The subject of the study is timely and important to the understanding of the functions of mammalian replicases.

However, there are problems with the paper, including inconsistent results, incomplete methods description, and overstatements. These points should be addressed before publication.

See specific comments below.

1. Page 3, the authors state: "... other residues within the FeS cluster binding pocket can potentially stabilise the co-factor, e.g. histidine residue through hydrogen bonding (Bak & Elliott, 2013)". However, the paper the authors refer to (Bak & Elliott, 2013), indicates the opposite: "The mitoNEET [2Fe-2S] cluster demonstrates proton-coupled electron transfer (PCET) and marked cluster instability, which have both been linked to the single His ligand."

We thank the reviewer for this comment. We did indeed not cite the reference correctly. We have changed the sentence, as follows:

“In a number of FeS proteins it has been shown that – apart from the cluster-ligating cysteines – other residues within the FeS cluster binding pocket can potentially stabilise the co-factor, e.g. protonable residues through hydrogen bonding”

2.Data in Fig. 1c and Supplementary Fig. 1b indicate that there is a low, although measurable amount of iron bound to the C1076S. Accordingly, without measuring the background level of radioactivity the authors should not claim a complete loss of the Fe-S cluster in C1076S pol d.

We agree with the reviewer that the iron levels of the cysteine variants do not go down to zero. What we do know is that they reach levels that are close to what we measure with non-FeS proteins. To not overstate our data, we have changed our wording:

“Replacing one of the four invariant cysteines of the CysB motif leads to a nearly complete loss of the FeS cluster.”

3.Page 7, “...we hypothesize that the FeS cluster could potentially be required to control the conformational flexibility of the linker and consequently regulate the switch between DNA polymerization and proofreading.” In order to test this hypothesis,...” The conducted experiment (Fig. 7b, Supplementary Fig. 5) does not provide any information about the conformation of the flexible linker, or its role in the optimal positioning of the enzyme on the DNA substrate (another presumption of the authors) and in regulating of the switch between polymerase and exonuclease.

We agree with the reviewer that the wording was a bit awkward and have reformulated this paragraph:

“Based on the proximity of the FeS cluster to the flexible linker in our model, it seems conceivable that already small structural changes in the FeS cluster-binding pocket may be able to influence the conformational flexibility of the linker and – by doing so – affect the alignment of Pol delta on the DNA substrate, and possibly the balance between DNA polymerase and exonuclease activities.

To test whether the equilibrium between the two catalytic activities of Pol delta is affected by alterations in the FeS cluster-binding motif, we carried out fixed-time primer extension assays...”

4.It appears that there is more unreacted substrate in reactions with 20nM CS Pol d + PCNA and 100 μ M dNTPs (supplementary Fig. 5b), than with 10-fold less enzyme (2 nM CS +PCNA and 100 μ M dNTPs) at 5 min time point (Fig. 3c). Why is that? These results seem to be inconsistent.

We agree with the reviewer that these results appear inconsistent. However, they are not directly comparable, since the experiments were performed in a slightly different way.

The time-resolved primer extension assays (**Figure 2**) were performed without pre-incubation of Pol delta with the DNA substrate and in a reaction buffer containing 20 mM

KCl. In contrast, in the dNTP concentration-dependent exonuclease-to-polymerase switch experiments (**Figure 6**), Pol delta was pre-incubated with the DNA substrate and reactions were performed in a reaction buffer containing 30 mM KCl. Since pre-incubation should not decrease the reaction efficiency, we suspect that it is mainly the salt concentration that affects the reaction outcome. Perhaps the partial DNA binding defect of Pol delta-CS is even more pronounced at higher ionic strength, resulting in lower primer extension efficiency.

5.The fidelity measurements are difficult to evaluate due to the incomplete description of the treatment of the data, as well as the data presentation.

The mutant frequencies for the three independent determinations should be presented to allow evaluation of the reproducibility of the measurements.

Despite the fact that the method was previously published, it would be helpful if the authors included the formula for calculating the error rate in the legend to Table 1. Also, the value of the subtracted background mutation frequency for the pSJ4 plasmid-based substrate should be stated. Was the assay based on the pSJ4 plasmid published previously? If so, please add reference. If not, a more detailed description should be included.

It is not clear how the target size (145) used for calculating the error rates was determined. The authors state that the target size-the number of detectable sites- is for all types of possible mutations (base substitutions, deletions and insertions). However, supplementary Table 2 lists "mixed" under "mutation types". There is no explanation what errors are considered to be in the "mixed". It is not clear how the target size is determined for the "mixed" category.

We have modified the text of the manuscript and provide now additional information regarding the pSJ4 fidelity assay. The description of the tables showing the pSJ4 fidelity data was changed both in the main manuscript and in the supplementary section, allowing the reader now to rationalise how the error rates were calculated. We now also reference a previous publication, in which this method was successfully employed to measure the fidelity of human PrimPol (Guilliam *et al.*, 2014). Additionally, we have included a supplementary figure (**Figure S4**) to provide more information on the preparation of the gapped plasmid, the detectable mutation sites within the gap and the formula used for the calculations of error rates.

Minor points

5.Page5, "To investigate the influence of the Fe-S cluster on Pol d's proofreading abilities, we purified...." An enzymes exonuclease activity is not the same as its proofreading ability.

We agree with the reviewer and have corrected this sentence.

6.Legend to supplementary Fig. 4 should contain explanation of the annotations of the mutational spectra. blue)

We agree that the figure legend was not sufficiently clear and have added additional explanations.

7.The model in Fig. 7a and its description in the text would be easier to follow if the polymerase subdomains (palm, fingers, thumb) were color coded (e.g. different shades of blue).

We understand the reviewer's concern. However, given the size and the multi-domain nature of the structure, we used one single colour per domain for the over-all clarity of the domain presentation. We fear that using different shades of blue for the polymerase subdomains would be rather confusing and would therefore prefer to leave the colouring unchanged.

A wild-type Pol δ

B Pol δ -CS

Attachment 1. S200 gel-filtration fractions of Pol δ and Pol δ -CS purified in the presence of PCNA. (A) Wild-type Pol δ . (B) Pol δ -CS.

A**B**

P12004	PCNA_HUMAN	1	MFEARLIVQGSILKVKVLEALKDLINAEACWDISSSGVNLQSMDSHVSLVQLTLRSEGFDTY	60
Q0GYH3	Q0GYH3_9ABAC	1	MFEAEFKTGAVLKRVLVETFKDLLPHATFDCDNRGVSMQVMDTSHVALVSLQLHAEGFKKY	60
			****.: *::**:::***:.*:*. .. **.:* **::**:* **.*::**.*	
P12004	PCNA_HUMAN	61	RCDRNLAMGVNLTSMKILKCAINEDIITLRAEDNADTLALVFEAPNOEKVSDYEMKLM	120
Q0GYH3	Q0GYH3_9ABAC	61	RCDRNVTLNVSINSLSKIIVKCVNERSVLMKAEDQGDVMAFVFN--DNRICTYTLKLMC	118
			*****:::.*:::***:* * .. : :*****.*::**:*: : : : . * :***	
P12004	PCNA_HUMAN	121	LDVEQLGIPEQEYS CVVKMPSGEFARICRDLSHIGDAVVISCAKDGVKFSASGELGNGNI	180
Q0GYH3	Q0GYH3_9ABAC	119	IDVEHLGIPDSYDCVVHMSSVEFAQVCKDMTQFDHDIIVSCSKKGLQFRANGDIGSADV	178
			::**:* **:::.* **::**:* * **::**:*: : : : **:* **:* **:*	
P12004	PCNA_HUMAN	181	KLSQTSNVDKEEEAVTIEMNEPVQLTFALRYLNFFTKATPLSSTVTLSMSADVPLVVEYK	240
Q0GYH3	Q0GYH3_9ABAC	179	QMSADN-----ENFSVLKAKQTVTHTFAGDYLCHEFAKAAPLPTVTIYMSEELPFKLEYC	233
			::* . * : : : * ** * ** . * : ** : ** : ** : * * : : : * : **	
P12004	PCNA_HUMAN	241	IADMGHLKYYLAPKIEDEEGS--	261
Q0GYH3	Q0GYH3_9ABAC	234	IKDVGVLACFLAPKIVNNDEEIF	256
			* * : * * : ***** : : : .	

Attachment 2. Mass spectrometry analysis of SDS-PAGE band.

(A) Scaffold result of analysis. (B) Alignment of human and baculovirus PCNA.

A wild-type Pol δ

B Pol δ -CS

Attachment 3. S200 gel-filtration fractions of Pol δ and Pol δ -CS purified in the presence of PCNA after 3 minutes of heat stress. (A) Wild-type Pol δ . (B) Pol δ -CS.

Attachment 4. Test of Flag-PCNA for DNA polymerase (+ dNTPs) and exonuclease (- dNTPs) contaminations. Increasing amounts (50-400 nM) of FLAG-PCNA were incubated with 20 nM of primer-template substrate.

A**B**
Attachment 5. DNA synthesis by Pol δ and PCNA on gapped pSJ4-lacZ α in the presence or absence of yRFC and ATP. (A) Schematic of assay. (B) Agarose gel showing digestion products.

Attachment 6. Co-IP of Flag-POLD1 and untagged PCNA expressed in Sf9 insect cells.

June 19, 2019

RE: Life Science Alliance Manuscript #LSA-2019-00321-TR

Prof. Kerstin Gari
University of Zurich
Institute of Molecular Cancer Research
Winterthurerstrasse 190
Zurich, Zurich 8057
Switzerland

Dear Dr. Gari,

Thank you for submitting your revised manuscript entitled "Human DNA polymerase delta requires an iron-sulfur cluster for high-fidelity DNA synthesis". Your manuscript has been now re-assessed by the original reviewers.

As you will see, reviewer #1 is not satisfied with the revision as the example gel-filtration shows that a lot of PolD1 elutes in later fractions than the stoichiometric complex - implying that you are working throughout your biochemical assays with a mixture of complexed and uncomplexed PolD1. The reviewer therefore questions the model put forward in the last figure. We have discussed your work in light of these comments, also with reviewer #2. We think that the comparative analysis of wildtype and mutant Pol Delta remains valuable to the field. However, we think that reviewer #1's concern needs to get addressed, therefore:

- please include a fully annotated version of attachment 1 as supplementary information and change the manuscript text to provide a rationale for why you chose to omit the gel filtration step from the purification protocol to inform the reader about the presence of multiple Pol delta species in the samples
- please make sure that the text reflects that the model in figure 7 is speculative
- please address the last point raised by ref 1 concerning PCNA elution (high-molecular weight fraction containing both PolD1 and PCNA)

Furthermore,

- please address the remaining concerns of reviewer #2 and #3
- please upload the supplementary figures without legends and as individual figure files. - please include the supplementary figure legends and the supplementary tables in the main manuscript file

Once we receive such a further revised version, we'd be happy to proceed towards acceptance of your manuscript.

Sincerely,

Andrea Leibfried, PhD
Executive Editor
Life Science Alliance

Meyerhofstr. 1
69117 Heidelberg, Germany
t +49 6221 8891 502
e a.leibfried@life-science-alliance.org
www.life-science-alliance.org

A. FINAL FILES:

B. MANUSCRIPT ORGANIZATION AND FORMATTING:

Reviewer #1 (Comments to the Authors (Required)):

The manuscript has undergone a major revision since the first submission. Although it has improved, there still are questions that are unanswered. There are still many flaws with this manuscript, but I will here only lift the flaw that by itself give the most severe consequence for all downstream results and interpretations. In brief, this by itself, does not allow any type of model to be put forward in the discussion.

The authors have attached the results when passing the Flag-purified Pol Delta over a size exclusion column. I can still not find that experiment shown as a figure in the manuscript. Nevertheless, the author claim " We had initially purified the Pol delta variants with a two-step purification: Flag-PD and S200 gel filtration, as seen in attachment 1 for the wild-type and CS variant enzyme (purified in the presence of PCNA). While the complexes were stable and eluted stoichiometrically in a defined peak, their purity did not appear to be improved, as compared to a one-step Flag-pull-down purification. Given that the additional step had prolonged the procedure (which we feared may affect the stability of the FeS cluster) and led to a dilution of the samples without further purification, we had decided to abstain from a second purification step."

When I inspect attachment 1 I find that the Flag-PD of Pol delta does NOT purify as a stoichiometric complex. The proteins in attachment 1 is not labeled, but based on a comparison with the gel in figure 2A I conclude the following.

In the gelfiltration, PolD1 is likely to be the top band. This subunit elutes over a large number of fractions, but with a peak in fraction 41/42. The next dominant band is likely to be PolD2 and this subunit elutes in a more focused manner with a peak around fraction 34/35. The next major band should be PolD3, which elutes about one fraction after PolD2 with a peak at 35/36. This is best seen when looking at the shoulders on the gel. Then at the bottom is a dominant protein band that I interpret should be the band labelled with a star in Fig 2A. Based on the updated manuscript this is a contamination of baculoviral PCNA of about 27 kDa size. This protein peaks at fraction 31/32.

To summarize, the peak elution for all four proteins are at fraction 31/32, 34/35, 35/36, and 41/42. Inspecting shoulders of the elution of each protein makes it even more clear that the proteins elutes at different time points from the size-exclusion column. In conclusion, the Flag-PD Pol delta is not purified as a stoichiometric complex! The same conclusion is drawn when inspecting Flag-PD Pol delta- CS. In fact the results are almost identical.

As a result, all biochemical experiments are carried out with a mixture of sub-complexes with Pol delta. Based on the general line of discussion in the manuscript and the models toward the end, it is particularly serious to have a mixture of PolD1 by itself, PolD1+PolD2, and other combinations of

subcomplexes in all the assays. This will make it impossible to know which complex contributes to what result in the downstream analysis of all experiments and interpretations made by the authors.

Finally, I have a question; have the authors reflected over how it can be that Baculoviral PCNA elutes before Pol delta? PCNA is a trimer of 27 kDa with a globular shape, and thus elutes around 90 kDa from the gelfiltration column. Pol delta is much larger, above 200 kDa and with an elongated shape. Thus Pol Delta should elute much earlier than PCNA from the gelfiltration column.

Reviewer #2 (Comments to the Authors (Required)):

I have read the revised manuscript by Jozwiakowski and colleagues. I am satisfied that the authors have addressed the points raised in the original review, and that the manuscript is now suitable for publication.

I have the following remaining minor points, that will improve the readability of the manuscript:

Add a suitable reference or remove comment about alleged known behaviour of multi-subunit enzymes, that first dissociate into subunits and then unfold, upon heating. As far as I know, this is not an established behaviour.

Remove comment about Pol delta having two binding interfaces to the primer/template. It only has one specific interface, at the primer/template junction. It might still be able to bind to the ds- or ssDNA regions, but these are not specific interfaces.

Was the same ssDNA probe used in Fig.3C and D? They seem to have different mobilities in the gel.

Reviewer #3 (Comments to the Authors (Required)):

Overall the authors' revisions in response to my comments are satisfactory.

I have two additional points for the authors' consideration, but I do not need to see the paper again.

In the legend to Table S1, please correct what appears to be a typo in the definition of the mutation frequency, from "- background mutation rate" to "- background mutation frequency".

Furthermore, it is more accurate to refer to the ratio of "number of white colonies/total number of colonies" as mutant frequency rather than mutation frequency. The number of mutations in the lac Z target sequence, which depends on the accuracy of the studied polymerase, may not be equal to the number of white, mutant colonies. Low-fidelity polymerases may generate more than one mutation per target sequence.

**Universität
Zürich**^{UZH}

Institute of Molecular Cancer Research

University of Zurich
Winterthurerstrasse 190
CH-8057 Zürich
www.imcr.uzh.ch

Dr Andrea Leibfried
Life Science Alliance
Editorial Office
Meyerhofstrasse 1
D-69117 Heidelberg
Germany

Kerstin Gari, PhD
Assistant Professor
e-mail: gari@imcr.uzh.ch
phone: +41 (0) 44 635 3468
fax: +41 (0) 44 635 3484

Re: Submission of revised manuscript

Zurich, 26 June 2019

Dear Dr Leibfried,

We would like to submit a revised version of our manuscript entitled “Human DNA polymerase delta requires an iron-sulfur cluster for high-fidelity DNA synthesis”. We have followed your advice and included the reviewers’ comments as follows:

Reviewer 1:

- We have now included a fully annotated version of attachment 1 (size exclusion analysis of Pol delta) as Figure S2A, and discuss the presence of sub-complexes and aggregated proteins (mainly POLD1 and PCNA in high molecular weight void fractions). We also provide a rationale for why we chose to stick to a one-step purification (page 4).
- We now refer to our model as “hypothetical” and “speculate about a scenario” (page 8 and 18).

Reviewer 2:

- We have removed our comment that multi-subunit enzymes first dissociate into subunits and then unfold upon heating (page 5).
- We have rephrased the paragraph about Pol delta having two binding interfaces to the primer-template. We now do not talk about “interfaces” anymore but rather say that Pol delta contacts a primer-template substrate in the double-stranded DNA region via its C-terminus and the single-stranded region via its N-terminus (page 5/6).
- As for our ssDNA EMSA data – we did in fact use the same ssDNA probe in both Fig 3C and 3D (now Fig 3E and 3F), but did not run the gels exactly for the same amount of time. Since the percentage of these PAA gels is very low, even small differences in running time (in our case about 10 minutes) cause a visible change in migration.

Reviewer 3:

- We have corrected the legends to Table 1 and S1 (page 19).

We have also uploaded the supplementary figures without legends and as individual figure files, and included the supplementary figure legends and supplementary tables in the main manuscript file.

We thank you once again for your constructive comments on our manuscript, and hope you find that the manuscript is now suitable for publication in *Life Science Alliance*.

With our very best wishes,

Stanislaw K. Jozwiakowski, PhD

Kerstin Gari, PhD

June 27, 2019

RE: Life Science Alliance Manuscript #LSA-2019-00321-TRR

Prof. Kerstin Gari
University of Zurich
Institute of Molecular Cancer Research
Winterthurerstrasse 190
Zurich, Zurich 8057
Switzerland

Dear Dr. Gari,

Thank you for submitting your Research Article entitled "Human DNA polymerase delta requires an iron-sulfur cluster for high-fidelity DNA synthesis". I appreciate how you addressed the remaining concerns and it is a pleasure to let you know that your manuscript is now accepted for publication in Life Science Alliance. Congratulations on this interesting work.

DISTRIBUTION OF MATERIALS:

Again, congratulations on a very nice paper. I hope you found the review process to be constructive and are pleased with how the manuscript was handled editorially. We look forward to future exciting submissions from your lab.

Sincerely,

Andrea Leibfried, PhD
Executive Editor
Life Science Alliance
Meyerohofstr. 1
69117 Heidelberg, Germany
t +49 6221 8891 502
e a.leibfried@life-science-alliance.org
www.life-science-alliance.org